# Impact of whole-genome duplications on structural variant evolution in *Cochlearia*

Tuomas Hämälä [1,2] ✉, Christopher Moore [1], Laura Cowan[1], Matthew Carlile[1], David Gopaulchan[3], Marie K. Brandrud[4], Siri Birkeland[4,5], Matthew Loose[1], Filip Kolář[6,7], Marcus A. Koch [8] & Levi Yant [1,6] ✉

Polyploidy, the result of whole-genome duplication (WGD), is a major driver of eukaryote evolution. Yet WGDs are hugely disruptive mutations, and we still lack a clear understanding of their fitness consequences. Here, we study whether WGDs result in greater diversity of genomic structural variants (SVs) and how they influence evolutionary dynamics in a plant genus, *Cochlearia* (Brassicaceae). By using long-read sequencing and a graph-based pangenome, we find both negative and positive interactions between WGDs and SVs. Masking of recessive mutations due to WGDs leads to a progressive accumulation of deleterious SVs across four ploidal levels (from diploids to octoploids), likely reducing the adaptive potential of polyploid populations. However, we also discover putative benefits arising from SV accumulation, as more ploidy-specific SVs harbor signals of local adaptation in polyploids than in diploids. Together, our results suggest that SVs play diverse and contrasting roles in the evolutionary trajectories of young polyploids.

Whole-genome duplication (WGD) is a dramatic mutation that directly challenges the stability of meiosis and DNA management[1,2]. As such, WGDs are often fatal, but the resulting polyploids that survive these initial obstacles may ultimately thrive[3]. Indeed, WGDs have likely contributed to the emergence of all major eukaryotic lineages[4], with particular importance in the evolution of plants[5]. WGDs also have a direct economic impact, as the majority of our most important crop species are polyploid[6]. Understanding how evolutionary dynamics are altered by WGDs is, therefore, a fundamental goal in evolutionary biology, with applications reaching into agriculture. However, much of the genomic work related to WGDs is conducted on allopolyploids (polyploids resulting from the joining of two lineages), in which the effects of WGDs are confounded by hybridization. Polyploids resulting from within-species WGDs (autopolyploids), by contrast, allow decoupling of the effects of WGDs from those of hybridization, providing feasible systems to assess how evolutionary processes are shaped by WGDs.

Autopolyploidy is typically characterized by random pairing of chromosomes in meiosis (in allopolyploids chromosome pairing typically happens within subgenomes), resulting in predictable changes in population genetic processes[7,8]. All else being equal, doubling the genome increases the mutational input, number of recombination events per individual, and the effective population size, leading to an increase in genetic diversity and a decrease in effective linkage[9–12]. Dominance structure is also transformed by WGDs, leading to more efficient masking of recessive mutations[13,14]. Thus, increased diversity combined with masking of deleterious mutations may initially raise the adaptive potential of nascent polyploids[15]. In the long-term, however, the hidden deleterious mutations might prove difficult to purge, and allelic masking not only increases genetic load but also reduces the efficacy of positive selection[13,14,16,17], resulting in negative fitness consequences for aging polyploids[18]. We can, therefore, expect both beneficial and detrimental effects arising from WGDs, with empirical support found for some of the theoretical predictions[19–22].

[1]School of Life Sciences, University of Nottingham, Nottingham, UK. [2]Production Systems, Natural Resources Institute Finland, Jokioinen, Finland. [3]School of Biosciences, University of Nottingham, Nottingham, UK. [4]Natural History Museum, University of Oslo, Oslo, Norway. [5]Faculty of Chemistry, Biotechnology and Food Science, Norwegian University of Life Sciences, Ås, Norway. [6]Department of Botany, Faculty of Science, Charles University, Prague, Czech Republic. [7]Institute of Botany, Czech Academy of Sciences, Průhonice, Czech Republic. [8]Centre for Organismal Studies, University of Heidelberg, Heidelberg, Germany. ✉e-mail: tuomas.hamala@luke.fi; levi.yant@nottingham.ac.uk

Despite decades of work on polyploid genetics, the impact of WGDs on the abundance and composition of genomic structural variants (SVs) remains unknown. SVs encompass variants that influence the presence, abundance, location, and/or orientation of the nucleotide sequence, typically defined as being greater than 50 bp in length. Studies of diploid organisms have established that SVs generally cover much more of the genome than point mutations[23–26], suggesting that they can have a major influence on the adaptive potential of populations and species. Given their disruptive effects on chromosomal structure, newly emerged SVs tend to be strongly deleterious and thus reduce the fitness of the host[27,28]. Yet SVs have also been associated with adaptive phenotypes in multiple species[29–34], demonstrating that individual SVs can have beneficial fitness effects. In polyploids, however, the trajectory of SV evolution is poorly understood, with existing knowledge primarily coming from allopolyploid crop genomes[35–37]. In turn, we are missing an assessment of SV diversity in wild autopolyploid systems, leaving unknown the impact of WGDs on SV evolution in natural contexts. Given the increased mutational input in polyploids, combined with their more complicated recombination and DNA repair machinery[2], we may expect SV emergence to increase as a result of WGD. This hypothesis is supported by recent empirical work in both autopolyploid *Cochlearia officinalis*[38] and *Cardamine amara*[39], which point to the rapid evolution of DNA repair genes. These selective sweeps suggest an early 'mutator' phenotype that generates excess SVs before the adaptation of the repair machinery to the polyploid cell state[38].

Here, motivated by the earlier theoretical and empirical results, we first quantify SV diversity in recent autopolyploids and then explore the evolutionary impact of the shifted SV landscape. We specifically ask how SVs influence the genetic load of polyploid populations, but also explore whether SVs provide unique benefits to polyploids. By analyzing hundreds of genomes from the plant genus *Cochlearia* (Brassicaceae), we find both negative and positive interactions between WGDs and SVs. Masking of recessive mutations has increased the accumulation of deleterious SVs in polyploids, likely reducing the adaptive potential of these populations. However, we also discover apparent benefits resulting from the accumulation of SVs, as many more ploidy-specific SVs harbor signals of possible local adaptation in polyploids than in diploids. Finally, we propose that range-edge populations can especially benefit from large-effect SVs, and that SV-mediated adaptation could become more prominent in the future due to rapid climate change. Overall, our results provide important insights into the evolutionary relationship between WGDs and SVs – an aspect that likely has a major impact on the adaptive potential of polyploid organisms.

## Results

### Genetic composition of the *Cochlearia* genus

To study the impact of WGDs on SV evolution in wild species, we conducted extensive long- and short-read sequencing on the *Cochlearia* genus. *Cochlearia* represents a reticulate species complex with two-thirds of its 20 accepted taxa polyploid[40,41], mostly of allopolyploid origin[42,43]. Autopolyploids still comprise an important part of the genus, including a widespread and successful autotetraploid, *C. officinalis*[41,44]. The evolutionary history of the genus is highly affected by glaciation and deglaciation processes. Many species are adapted to cold and wet environments[45], reflecting the fact that *Cochlearia* expanded their distribution range northward during the Pleistocene, rapidly diversifying to new ecological conditions in central and northern Europe as well as across the circumarctic[40,41]. As an evolutionarily dynamic genus, *Cochlearia* exhibits a highly labile genome structure, with two base chromosome numbers ($x = 6$ and $x = 7$) and multiple ploidal levels (from diploids to dodecaploids) found among the species[40,42,43].

Here, we focus on populations from the diploid $x = 6$ species *C. pyrenaica*, *C. excelsa*, *C. aestuaria*, and *C. islandica*; diploid $x = 7$ species *C. groenlandica* and *C. triactylites*; tetraploid $x = 6$ species *C. officinalis* and *C. alpina*; tetraploid $x = 7$ species *C. micacea*; hexaploid $x = 6$ species *C. bavarica* and *C. polonica*; hexaploid $x = 7$ species *C. tatrae*; and octoploid $x = 6$ species *C. anglica*. The tetraploids likely resulted from within-species WGDs (autopolyploids), as evidenced by widespread multivalent formation at meiosis[38], whereas the evolutionary history of the higher ploidies is more complex, involving both auto- and allopolyploidization events. The hexaploid *C. tatrae*, *C. bavarica*, and *C. polonica* are locally distributed endemics from very different habitats in Europe and likely evolved independently from hybridization between diploid *C. pyrenaica* and differing sub-gene pools of tetraploid *C. officinalis*. The octoploid *C. anglica* most likely evolved from a second autopolyploidization event of *C. officinalis*. See Koch[40] and Wolf et al.[41] for more information about the evolutionary history of the species as well as an extensive systematic and taxonomic survey of the *Cochlearia* genus.

In total, our dataset comprised 23 samples sequenced with Oxford Nanopore (ONT) or Pacific Biosciences (PacBio) long-read technologies and 351 samples sequenced with Illumina short-read technology. The individuals represent 76 populations, covering the primary range of *Cochlearia* throughout Europe (Fig. 1A), along with locations in Svalbard and North America (Dataset S1). We first used SNP data derived from short-read sequencing to examine patterns of genetic diversity and differentiation among the *Cochlearia* populations. Compared to the diploids, polyploid populations exhibited lower levels of genetic diversity (Fig. 1B) and more negative Tajima's $D$ (Fig. 1C), potentially reflecting bottlenecks and subsequent expansions resulting from the recent establishment of these populations[41]. A principal component analysis (PCA) indicated genetic clustering primarily due to geographical location: the first two principal components (PC) corresponded to multiple locations, while also revealing some separation due to ploidy (Fig. 1D). The geographical clustering was also evident in within-ploidy PCAs, while little separation was found based on species assignments (Supplementary Fig. 1). We further discovered a signal of isolation-by-distance, with between-population $F_{ST}$ estimates increasing with geographical distance, especially among the diploids (Fig. 1E). However, by using redundancy analysis (RDA) to model the role of geography, climate, and ploidy in explaining differentiation among the populations, we found climatic conditions to be a better predictor of genetic differentiation than either geographical distance or ploidy (Fig. 1F).

### SV identification and methylation assessment using long-read sequencing

Based on the analysis of SNP data, we found indications that polyploidy influences the genetic composition of the *Cochlearia* genus. To explore whether WGDs also have an impact on SV landscapes, we performed long-read sequencing to identify SVs in 23 samples chosen to represent diverse lineages and ploidies. However, due to low sequencing depth, we excluded four diploids from our main analyses (Supplementary Table 1), resulting in a set of 10 diploids, seven tetraploids, one hexaploid, and one octoploid. After aligning reads against the chromosome-build *C. excelsa* reference genome[38], we used Sniffles2[46] to identify SVs from the alignments. First, as Sniffles2 was developed primarily for diploid organisms, we used simulated data to confirm that it likely has good power to detect SVs in our high-depth (mean depth = 68) autotetraploid samples (Fig. 2A and Supplementary Table 2). We focused our analyses on insertions and deletions between 50 bp and 100 kb in size and filtered them for variant quality, missing data, and sequencing depth. After filtering, we retained 78,450 SVs in diploids and 111,363 in tetraploids. As both sequencing depth and read length can influence the power to detect

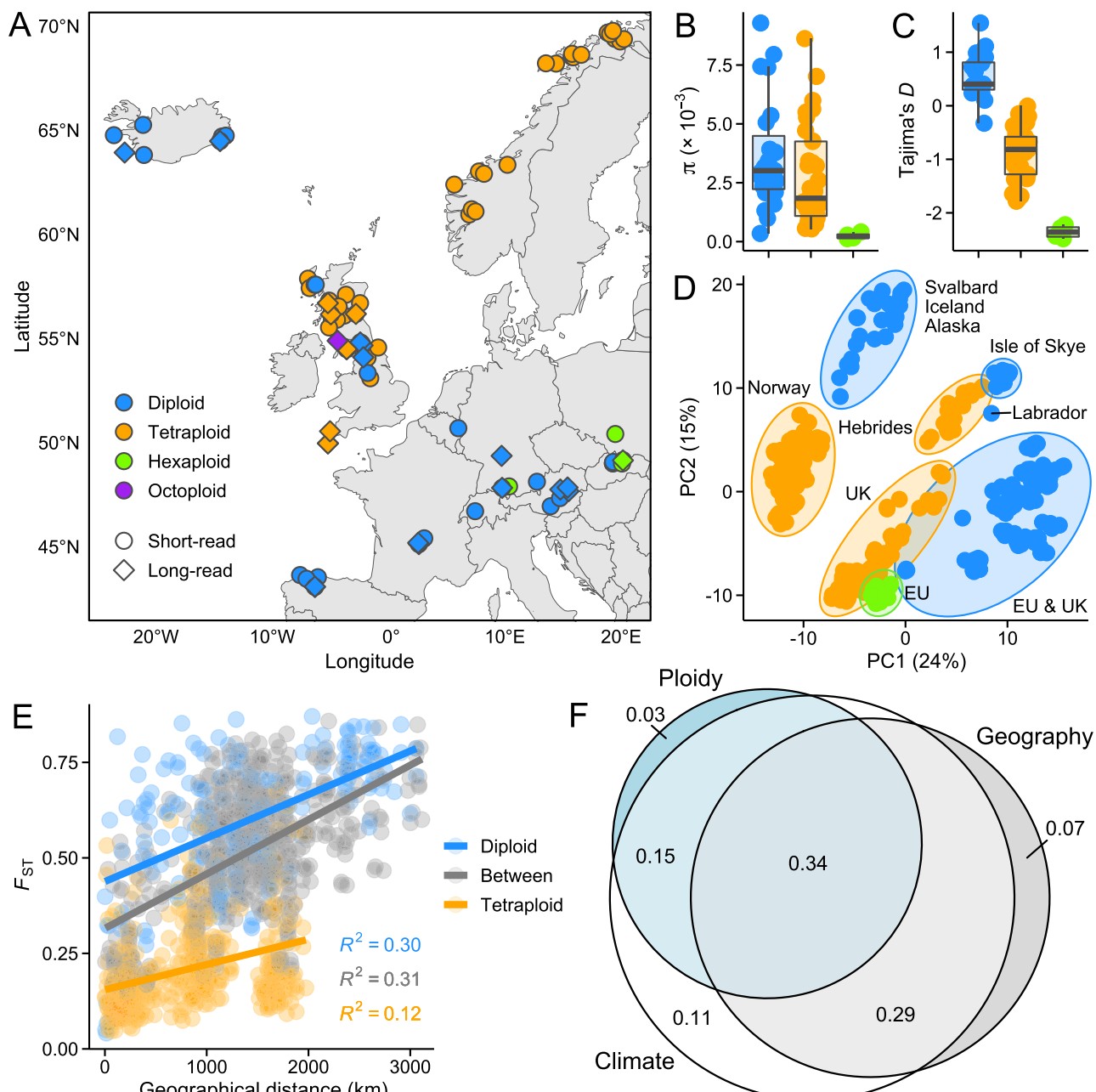

**Fig. 1 | Locations and genetic variation among *Cochlearia* populations used in this study. A** Map depicting European sampling locations. Shown are short-read sequenced populations (circles) and long-read sequenced individuals (diamonds). **B** Pairwise nucleotide diversity (π) estimates for the short-read sequenced populations with sample size ≥ 4 ($n_{diploid} = 23$, $n_{tetraploid} = 33$, $n_{hexaploid} = 4$). Center line, median; box limits, upper and lower quartiles; whiskers, 1.5 × interquartile range. **C** Tajima's D estimates for the short-read sequenced populations with sample size ≥ 4 (diploid $n = 23$, tetraploid $n = 33$, hexaploid $n = 4$). Center line, median; box limits, upper and lower quartiles; whiskers, 1.5 × interquartile range. **D** First two

axes of a principal components analysis (PCA). The proportion of variance explained by the principal components (PCs) is shown in parentheses. **E** Relationship between $F_{ST}$ and geographical distance among diploid and tetraploid populations (between = diploid vs. tetraploid). **F** The role of geography, climate, and ploidy in explaining genetic differentiation among these *Cochlearia* populations. Adjusted $R^2$ values from partial RDA models are shown in the circles. Note that the same color legend applies to panels **A**–**E**. Source data are provided as a Source Data file.

SVs, we confirmed that tetraploids also carried more SVs after downsampling the alignments to an equal number of base pairs (Supplementary Fig. 2). By comparing the SV sequences against our transposable element (TE) library, we found that in both diploids and tetraploids ~60% of the SVs contained TE sequence (Supplementary Fig. 3), suggesting that many of the SVs are likely the result of TE mobilization. To examine whether TE activity, and thus the potential of TEs to generate new SVs, differs between the ploidies, we quantified TE methylation using our ONT-sequenced samples.

Although we observed higher methylation levels in tetraploids than in diploids (Supplementary Fig. 4), the pattern was not unique to TEs, and once we controlled for the genome-wide difference in methylation levels, we saw no evidence that TE families are systematically hyper- or hypomethylated in tetraploids (Supplementary Fig. 5). Indeed, by estimating putative insertion times for TEs, we found no significant differences between the ploidies (Supplementary Fig. 6), indicating that differential silencing of TEs is not a major factor shaping SV landscapes in diploids and tetraploids.

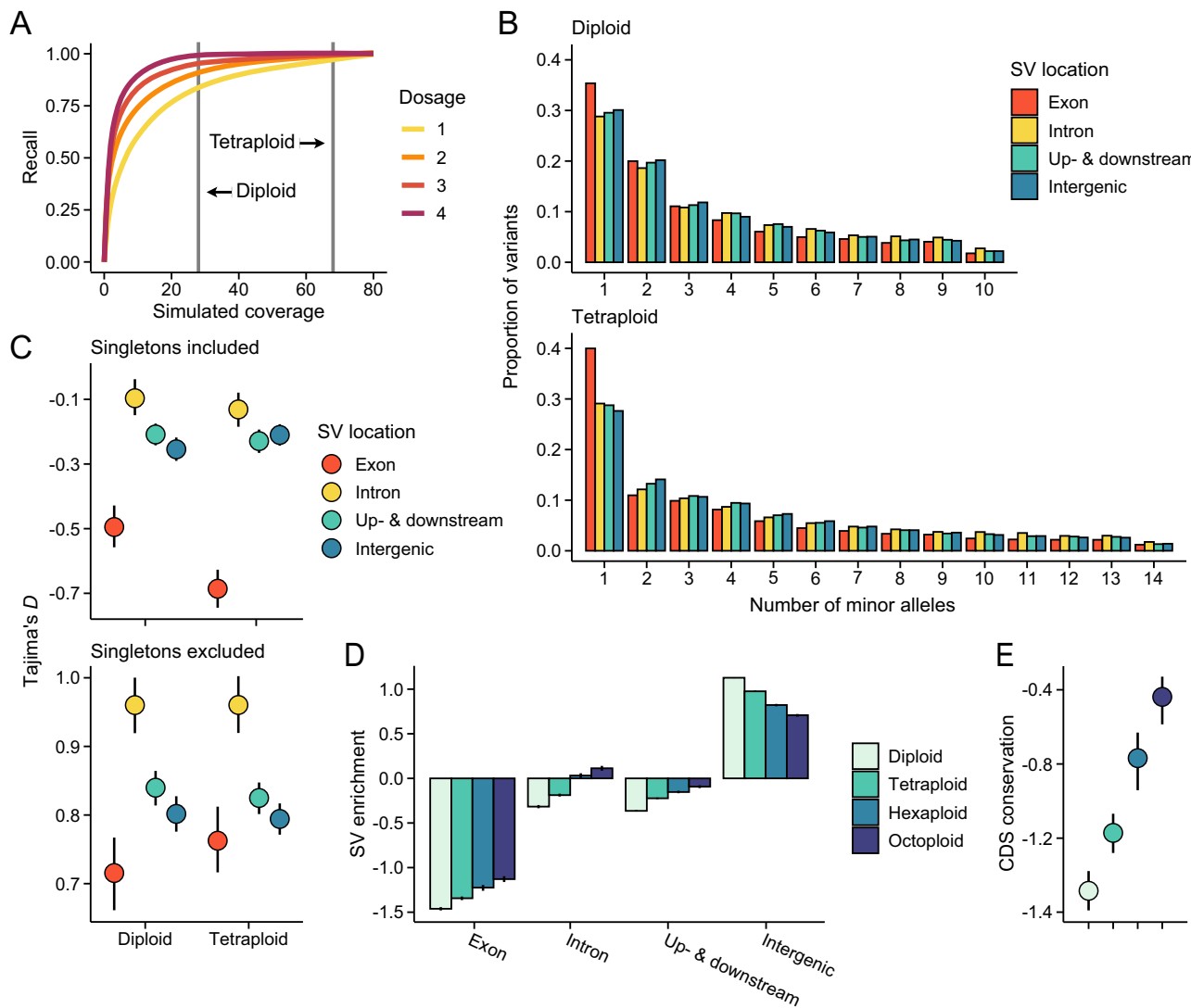

**Fig. 2 | Fitness effects of structural variants (SVs). A** Validation of the SV caller. Shown are recall estimates for different levels of read depth over simulated insertions and deletions. Grey horizontal lines mark the mean sequencing depth of our diploid and tetraploid samples. **B** Folded allele frequency spectra (AFS) for SVs found overlapping different genomic features in the long-read sequenced diploid and tetraploid samples. **C** Tajima's *D* estimated from the whole AFS (singletons included) and AFS with singletons excluded. **D** SV enrichment across different genomic features in diploids and polyploids. Shown are log$_2$-transformed ratios of observed to expected numbers of SVs. **E** Coding sequence (CDS) conservation of genes affected by SVs (exon in panel **D**). Shown are median standardized GERP scores estimated among 30 eudicot species (lower values indicate weaker conservation). Note that the same color legend applies to panels (**D**) and (**E**). In panels (**C**), (**D**), and (**E**) error bars indicate 95% bootstrap-based CIs. Source data are provided as a Source Data file.

## Masking progressively increases SV load in polyploids

To gain insight into the fitness effects of SVs, we estimated allele frequency spectra (AFS) for SVs found in exons and compared these to SVs overlapping regions less likely to have functional roles (introns, 1 kb up and downstream of genes, intergenic). Although we can expect that SVs found in the intergenic space are least likely to influence fitness, our simulations suggest that SV calls in such regions may suffer from excessive rates of false positives (~40%, Supplementary Table 2), likely due to the high density of repeats. Genic regions (≤1 kb), by contrast, had low false positive rates (~2%) regardless of the elements with which the SVs overlapped (Supplementary Table 2). By comparing the different SV classes between the ploidies, we found the most prominent difference to be an excess of rare exonic SVs in tetraploids (Fig. 2B). This pattern was confirmed by summarizing the AFS using Tajima's *D*: exonic SVs were segregating at lower frequencies in tetraploids than in diploids (Fig. 2C), whereas no substantial differences were found among the other SV classes (overlap between 95% CIs, Fig. 2C). In the absence of mutation rate difference, such excess could

either indicate stronger purifying selection in tetraploids or that recently emerged SVs are tolerated at functional regions because their effects are being masked by the additional allelic copies. To answer this question, we compared Tajima's *D* estimated from the whole AFS to estimates acquired after excluding singletons (i.e., variants with only a single allele present). We found that the exclusion of singletons removed the excess of rare exonic SVs in tetraploids (Fig. 2C), supporting the idea that such SVs are being retained due to more efficient masking (as stronger purifying selection would skew the whole AFS towards rare variants). Therefore, our AFS-based analyses suggest that masking of recessive mutations allows SVs to accumulate in tetraploids that would have been purged by purifying selection in diploids. We acknowledge, however, that these analyses rely on the correct identification of the SV genotypes, which can be challenging in polyploids, despite our validation (Fig. 2A). Thus, as an alternative approach, we examined the genomic locations of the SVs (regardless of their genotypes) and compared the observed numbers of SVs found overlapping different genomic features to random expectations. Given that this

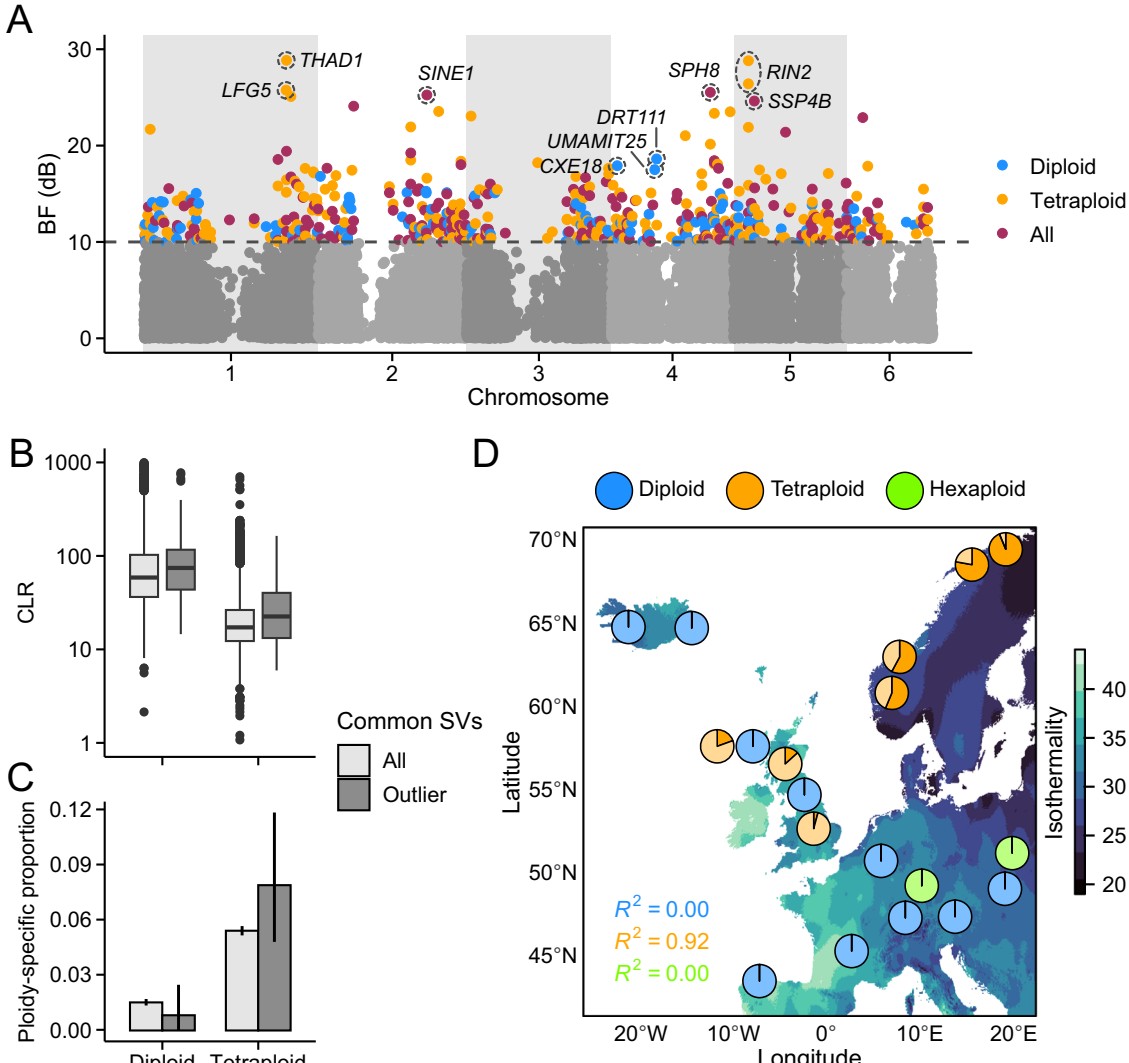

**Fig. 3 | Footprints of environmental adaptation at SVs. A** Bayes Factor (BF) estimates from genotype-environment association (GEA) analyses conducted on short-read sequenced diploid, tetraploid, and all (diploids, tetraploids, and hexaploids) populations. SVs with BF (dB) ≥ 10 were considered putatively adaptive. The top three candidate genes (< 1 kb of the SVs) from each analysis are highlighted. **B** Composite likelihood ratio (CLR) test statistics from sweep analyses conducted using SNPs ≤ 20 kb from each common SV (MAF > 0.05). $n_{\text{diploid}}$ $_{\text{all}}$ = 16,910; $n_{\text{diploid outlier}}$ = 124; $n_{\text{tetraploid all}}$ = 17,832; $n_{\text{tetraploid outlier}}$ = 234. Center line, median; box limits, upper and lower quartiles; whiskers, 1.5 × interquartile range; points, outliers. **C** The proportion of common SVs found only among diploid or tetraploid populations (ploidy-specific). $n_{\text{diploid all}}$ = 18,997; $n_{\text{diploid outlier}}$ = 124; $n_{\text{tetraploid all}}$ = 32,084; $n_{\text{tetraploid outlier}}$ = 234. Error bars show 95% bootstrap-based CIs. **D** Example of a tetraploid-specific outlier SV, found 300 bp upstream of a gene *RIN2*. Pie charts show the frequencies of reference (lighter color) and alternative (darker color) alleles in closely adjacent populations. Source data are provided as a Source Data file.

analysis does not require population-level data, we also included the hexaploid and octoploid samples. As expected, we discovered an overall deficit of exonic SVs and an excess of intergenic SVs (Fig. 2D). However, the deficit was greater in diploids than in polyploids, with the amount decreasing progressively with increasing ploidy (Fig. 2D). Furthermore, by examining the level of coding sequence conservation at genes affected by the SVs, we found a similar cline between all ploidies (Fig. 2E), indicating that SVs are being retained in genes under stronger selective constraint in polyploids than in diploids. Both results further support our conclusion that masking allows recessive SVs to accumulate in polyploids, likely progressively increasing the genetic load of higher ploidy populations.

### *Cochlearia* pangenome reveals climate-associated SVs

Although our analyses of the long-read data suggest that masking has increased the accumulation of deleterious SVs in polyploids, we might expect that some SVs provide selective benefits for the *Cochlearia*

populations. Therefore, to examine the potential role of SVs in environmental adaptation, we constructed a graph-based pangenome for *Cochlearia* and used it to genotype 257,807 SVs in 351 short-read sequenced samples. Using simulations, we first confirmed that this genotyping approach is well-suited for polyploid samples (Supplementary Table 3). After filtering the SVs for variant quality, missing data, and minor allele frequency (MAF), we used 18,997 (diploids), 32,084 (tetraploids), and 27,515 (all: diploids, tetraploids, and hexaploids) SVs to conduct genotype-environment association (GEA) analyses. Our analyses identified 124 SVs strongly associated with climatic variables in diploids, 234 in tetraploids, and 201 when considering all ploidal levels (Fig. 3A). To assess whether these SVs have been subject to recent positive selection in some of the *Cochlearia* populations, as might be expected if they are involved in adaptation to local environments, we searched for footprints of selective sweeps on SNPs likely in linkage with the SVs (≤20 kb from the breakpoints). Overall, composite likelihood ratio (CLR) test statistics were positively

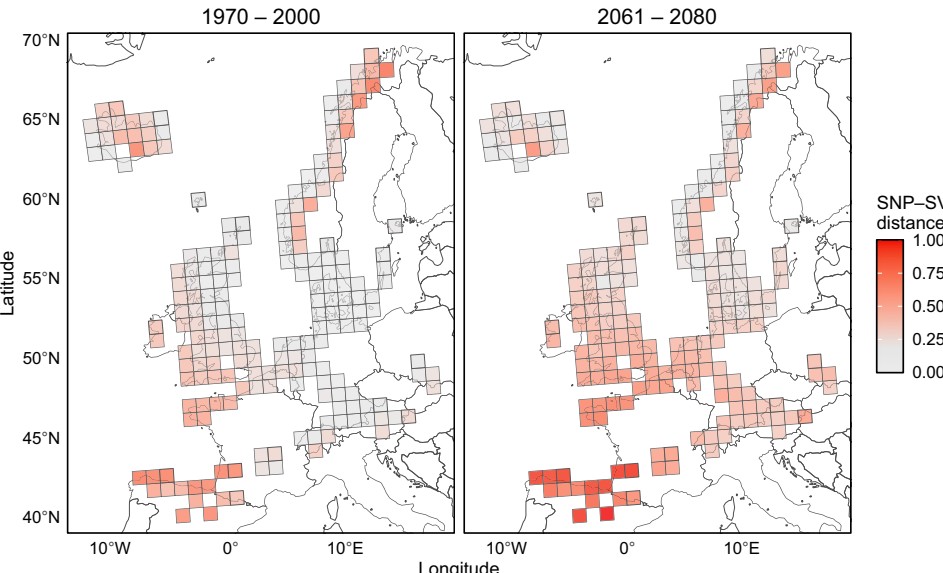

**Fig. 4 | Distance between climate-associated SNPs and SVs across the European range of *Cochlearia* species included in this study.** Color scale indicates the level of unique contributions that SVs are predicted to make to environmental adaptation (normalized Euclidean distance between the climatic indices). The first panel shows past adaptation based on 11 bioclimatic variables collected between 1970 and 2000, and the second panel shows a projection for the years 2061– 2080. Map data: GISCO, licensed under CC by 4.0. Source data are provided as a Source Data file.

correlated with Bayes Factor estimates from our GEA analyses (Spearman's $\rho = 0.42$, $P < 2 \times 10^{-16}$), suggesting that SVs with stronger association with climate are more likely to be affected by positive selection. Supporting this notion, we found more pronounced sweep signals among the outlier than non-outlier SVs in both diploid and tetraploid populations (Fig. 3B; $P < 0.01$, two-sided Wilcoxon rank-sum test).

Given the greater accumulation of SVs in polyploids, we might assume that ploidy-specific SVs are more likely to contribute to environmental adaptation in tetraploids than in diploids. Consistent with this expectation, we discovered that a larger proportion of common SVs (MAF > 0.05) were ploidy-specific in tetraploids than in diploids (Fig. 3C; $P < 2 \times 10^{-16}$, two-sided Fisher's exact test), including GEA outlier SVs (Fig. 3C; $P = 0.005$, two-sided Fisher's exact test). For example, two top outlier SVs in our GEA analyses, closely adjacent to the gene *RIN2*, were only polymorphic among the tetraploid populations (Fig. 3D). We further note that the proportion of ploidy-specific SVs in tetraploids is likely underestimated, as our long-read data do not cover the entire tetraploid range (a cluster of diversity in Norway is missing, Fig. 1D), whereas among diploids there was a close correspondence between long- and short-read sequencing (Fig. 1A).

Among the top outliers in diploids, we discovered SVs closely adjacent (<1 kb) to genes *DRT111* and *UMAMIT25*, involved in seed development and germination[47,48]; and *CXE18*, involved in pollen tube growth[49]. In tetraploids, the top candidate genes included *RIN2* and *LFG5*, involved in pathogen resistance[50,51]; and *THAD1*, involved in root development and gravitropism[52]. In the analysis comprising all ploidal levels, the top candidate genes were *SPH8*, involved in pollen-pistil interactions[53]; *SINE1*, involved in nuclear migration[54]; and *SSP4B*, involved in protein dephosphorylation[55] (Fig. 3A). Gene ontology (GO) terms related to germination (maintenance of seed dormancy) and pathogen resistance (hypersensitive response) were also enriched among all outliers in diploids and tetraploids, respectively (Supplementary Table 4). Interestingly, the candidate genes and biological processes associated with the outlier SVs were largely independent of those identified with SNPs, as only 37% of genes and none of the GO terms were represented in the SNP-based analyses.

## The role of SVs in environmental adaptation may increase due to climate change

Our results suggest that SVs could contribute to environmental adaptation in *Cochlearia*. To gain more insight into the geographical distribution of this climate-associated variation, we predicted climatic landscapes across the European range of our focal *Cochlearia* species (Supplementary Fig. 7). By leveraging the associations between genetic and environmental variables, climatic landscapes can be used to project climate-associated variation to unsampled locations[56] and to model population vulnerability under climate change[57]. Here, however, we extend this approach to identify geographical regions where SVs potentially make unique contributions to environmental adaptation by visualizing the difference between SV- and SNP-based landscapes. Our analysis identified the northern (Norway and Iceland) and southern (Spain and France) edges of the *Cochlearia* range as locations where the climatic landscapes are most strongly diverged (Fig. 4). Furthermore, by conducting a prediction using environmental variables projected for years 2061–2080, we discovered that the disparity between SV- and SNP-based climatic landscapes may increase due to climate change (assuming that populations mainly track climate change through existing variation), especially in populations at the southern edge of the *Cochlearia* range (Fig. 4). We note that the same populations are not, according to our analysis, the ones most vulnerable to climate change, which are primarily found at the eastern edge of the *Cochlearia* range (Supplementary Fig. 8).

## Discussion

How WGDs influence adaptive evolution is a long-standing question in evolutionary biology. Based on both theoretical[7,8,14,16] and empirical[19–22] work, we can expect pervasive fitness consequences arising from WGDs. However, the evolutionary relationship between WGDs and SVs is poorly understood, partly because SV identification has been challenging using short-read sequencing technologies[58].

Here, we used long-read sequencing and pangenomics to study the impact of WGDs on SV landscapes in the plant genus *Cochlearia*. We discovered a substantial accumulation of genic SVs in polyploids that likely would have been purged by purifying selection in diploids. Theory suggests that such hidden load can have a major impact on the

long-term fate of polyploid populations[13,14,17], contributing to eventual extinction or rediploidization[18]. Although previous studies have discovered an excess of nonsynonymous SNPs[21] and TEs[22] in recently founded autotetraploids, we may expect that SV accumulation has a particularly strong effect on the genetic load of polyploid populations. SVs are not only more deleterious than point mutations[27,28] but also could be more frequently generated in polyploids due to more complicated recombination and DNA repair machinery[2], as experimentally shown in yeast[19]. Indeed, we previously showed that genes involved in DNA repair have evolved rapidly in the tetraploid *C. officinalis* since its origin from the diploid *C. pyrenaica*[38], suggesting that WGD in *Cochlearia* has resulted in a shift in the (internal) selective environment due to extra challenges in DNA management.

Assuming that most deleterious mutations are partially recessive[59], SVs could have two major consequences for the fate of autopolyploid populations: 1) The point at which the genetic load of a newly formed autotetraploid population exceeds that of its diploid progenitor is reached faster with stronger selection coefficients[14], meaning that SVs (compared to point mutations) could shorten the period of beneficial fitness effects arising from WGDs. 2) Once a population reaches equilibrium, the fitness reduction due to deleterious mutations is roughly equal to the product of ploidal level and mutation rate[14,17], indicating that a higher rate of SV emergence in polyploids would increase the genetic load beyond that predicted from ploidy alone. Therefore, the accumulating SV load is likely an important factor limiting the adaptive potential of polyploid organisms, especially among the higher ploidies. We acknowledge, however, that in our hexaploid sample, the load inference could be influenced by its mixed auto and allopolyploid history[41], as subgenome dominance and lack of homoeologous recombination may increase the accumulation of deleterious mutations in allopolyploids compared to autopolyploids[60]. Nevertheless, the progressive accumulation of genic SVs across four ploidal levels supports the idea that increasing ploidy leads to more efficient masking of recessive mutations, thus reducing the efficacy of purifying selection.

Despite the increased SV loads in polyploids, we also discovered apparent benefits resulting from the SV accumulation. Among the climate-associated SVs, we found many more ploidy-specific variants in tetraploids than in diploids. Although functional validation of the detected SVs is beyond the scope of this study, their putative involvement in environmental adaptation suggests that the greater SV diversity in polyploids occasionally gets harnessed by positive selection. Furthermore, as interploidy gene flow is almost exclusively unidirectional from diploids to tetraploids[18,61], tetraploids are more likely to benefit from adaptive SVs originating in diploids than vice versa. Therefore, our results suggest that SVs contribute to the greater diversity of adaptive alleles available for polyploids[62], compensating for some of the detrimental effects arising from the increased SV load. By analyzing genes closely adjacent to the outlier SVs, we discovered enrichment of genes involved in different biological processes. The most prominent were related to seed germination in diploids and pathogen resistance in tetraploids – processes that were also associated with the top outlier genes from the corresponding GEA analyses. Importantly, the majority of the candidate genes and biological processes were not detected using SNPs, demonstrating that SVs need to be considered for a comprehensive view of adaptive processes.

To gain more insight into the unique roles of climate-associated SVs, we searched for differences between SV- and SNP-based climatic landscapes[63]. The northern and southern range edges were highlighted as regions where the climatic landscapes are most strongly diverged, potentially indicating greater contributions made by SVs to environmental adaptation. Indeed, we might expect to find more adaptive SVs in range-edge populations, as large-effect mutations tend to be favored in populations that are far from their selective optima[64,65]. By conducting a prediction using future climate projections, our modeling further suggests that the divergence between the SV- and SNP-based climatic landscapes may grow in the future, potentially as a result of SVs currently conferring adaptation to the southern environment increasing in frequency and spreading northward due to climate change. Furthermore, this analysis expects that populations track the shifting fitness optima through existing variation, but SV emergence could also increase due to climate change, as environmental stress is known to induce TE mobilization[66,67], potentially providing more opportunities for SV-mediated adaptation.

By conducting extensive long- and short-read sequencing on samples of varying ploidy (between diploid and octoploid) from the plant genus *Cochlearia*, we have gained important insights into the evolutionary relationship between WGDs and SVs. We discovered a progressive accumulation of genic SVs across four ploidal levels, indicating increased SV loads in polyploids compared to diploids. Given the strongly negative fitness effects of SVs, we expect such SV loads to limit the long-term adaptability of polyploid populations and species. However, by constructing a graph-based pangenome for *Cochlearia*, we also found putative benefits arising from the SV accumulation, as ploidy-specific SVs were more likely to harbor signals of local adaptation in tetraploids than in diploids. Finally, our modeling work highlighted the potential roles of SVs in adaptation to past and future climates. Overall, our analysis of SVs in *Cochlearia* sheds light on important but understudied aspects of polyploid genomes, broadening the perspective of polyploid evolution as well as the evolution of structural variation in wild populations and species.

## Methods

### Sampling
All samples were collected in compliance with local, national, and international laws in the following countries: Austria, Belgium, England, France, Germany, Iceland, Norway, Scotland, Slovakia, Spain, and Switzerland. Material from collections under curation/international exchange of Heidelberg botanical collections and herbarium was sourced between 2004 and 2022. Where applicable and relevant, we received permissions from Nagoya focal points in each country and submitted the Due Diligence Declaration to our relevant Competent Authority. A sampling of young leaf material into desiccant was performed in the field, aiming for at least 10 plants per population, with each sampled plant a minimum of two meters from any other. Collection dates and locations are detailed for all samples in the ENA archive at EMBL-EBI under accession number PRJEB66308. Geographic coordinates are also given in Dataset S1.

### High molecular weight DNA isolation, Oxford Nanopore, and PacBio HiFi sequencing
To study the evolutionary role of SVs in *Cochlearia*, we collected samples from 23 individuals to be used in long-read sequencing. The set included 14 diploids, seven tetraploids, one hexaploid, and one octoploid (Supplementary Table 1). Before starting DNA isolation, 20 mL of Carlson lysis buffer (100 mM Tris-HCl, 2% CTAB, 1.4 M NaCl, 20 mM EDTA, 1% PEG 8000) was mixed with 0.3 g PVPP and 50 μL B-mercaptoethanol and preheated to 65 °C. Leaf material from individual plants was ground into the heated solution and incubated for an hour at 65 °C. 20 mL chloroform was then added and mixed by inverting. The mixture was centrifuged at $3500 \times g$ (4 °C) for 15 minutes, the top layer of the lysate added to $1 \times$ volume isopropanol, inverted to mix, and incubated at −80 °C for 15 minutes before being centrifuged at $3500 \times g$ (4 °C) for 45 minutes. The supernatant was removed, the pellet air dried (sterile wipes were also used to dry the side walls of the tube) and resuspended in 500 μL nuclease-free water containing 2 μL of RNase A before being left to incubate at 37 °C for 45 minutes. Samples were column purified with a Qiagen Blood and Cell Culture DNA Maxi Kit clean up using 100/G columns. The DNA concentration was checked on a Qubit Fluorometer 2.0 (Invitrogen)

using the Qubit dsDNA HS Assay kit. Fragment sizes were assessed using the Genomic DNA Tapestation assay (Agilent). Removal of short DNA fragments and final purification to high molecular weight DNA was performed with the Circulomics Short Read Eliminator XS kit. After DNA isolation, two samples were used for Pacific Biosciences (PacBio) HiFi sequencing and 21 samples for Oxford Nanopore Technologies (ONT) sequencing.

ONT libraries were prepared using the Genomic DNA Ligation kit SQK-LSK109 following the manufacturer's procedure. Libraries were loaded onto R9.4.1 PromethION Flow Cells and run on a PromethION Beta sequencer. Due to the rapid accumulation of blocked flow cell pores or due to apparent read length anomalies on some *Cochlearia* runs, flow cells used in the runs were treated with a nuclease flush to digest blocking DNA fragments before loading with fresh libraries according to the ONT Nuclease Flush protocol (version NFL_9076_v109_revD_08Oct2018). FAST5 sequences produced by PromethION sequencer were basecalled using the Guppy6 (https://community.nanoporetech.com) high accuracy basecalling model (dna_r9.4.1_450bps_hac.cfg) and the resulting FASTQ files quality filtered by the basecaller. PacBio sequencing was performed on a Sequel IIe at Novogene Europe (Cambridge, UK) in CCS mode.

### Short-read library preparation and sequencing

We also used a set of 109 short-read sequenced *Cochlearia* individuals from Bray et al.[38], which includes 39 diploids and 70 tetraploids. Although this sampling covers several locations across Western and Northern Europe, it is mainly focused on the UK. To expand our sampling to more varied environments, we additionally collected 242 *Cochlearia* individuals across Europe and North America, leading to a final set of 351 individuals from 76 populations used for short-read sequencing. These samples comprise 148 diploids, 179 tetraploids, and 24 hexaploids (Dataset S1).

DNA was prepared using the commercially available DNeasy Plant Mini Kit from Qiagen (Qiagen: 69204). Illumina libraries were constructed from genomic DNA using the Illumina DNA Prep library kit and IDT for Illumina DNA/RNA Unique Dual Index sets. Library preparation was performed using a Mosquito HV (SPT Labtech) liquid-handling robot. The standard protocol timings and reagents were used but with 1/10th reagent volumes at all steps. A total of 9–48 ng of DNA was used as library input and 5 cycles of PCR were used for the library amplification step. Individual libraries were pooled together and size selected using 0.65 × AMPure XP beads to minimize library fragments <300 bp. Library pools were sequenced on a Novaseq 6000 using 2 × 150 bp paired-end reads at Novogene Europe (Cambridge, UK).

### Transposable element annotation

We previously identified TEs from the *C. excelsa* reference genome[38]. However, as the reference originated from a selfing diploid, we additionally assembled the genomes of one outcrossing diploid (*C. pyrenaica*) and one outcrossing tetraploid (*C. officinalis*) to expand our library of *Cochlearia* TEs. To do so, the individuals were sequenced using PacBio HiFi reads to an estimated depth of ~20 (diploid) and ~40 (tetraploid) × the haploid genome size. The reads were then de novo assembled using hifiasm[68] and haplotigs removed from the primary assemblies using purge_dups[69]. The resulting assemblies had a total size of 359 (diploid) and 315 (tetraploid) mb, with contig N50 of 2.6 mb (diploid) and 630 kb (tetraploid). BUSCO[70] analysis indicated high completeness of the gene space, with 96% of the single-copy Brassicales genes found in both assemblies (Supplementary Fig. 9). As with the *C. excelsa* reference genome, we annotated the assemblies using the EDTA pipeline[71], which includes multiple methods to comprehensively identify both retrotransposons and DNA transposons. To generate a single TE library across the three species, we used the cleanup_nested.pl script from EDTA to remove redundant (>95%

identical) consensus sequences from the combined library. We last conducted BLAST queries against a curated plant protein database from Swiss-Prot to remove likely gene sequences from the TE library. See Supplementary Fig. 10 for an outline of the annotated TE superfamilies.

### Short-read processing and SNP calling

Low-quality reads and sequencing adaptors were removed using Trimmomatic[72] and the surviving reads aligned to the *C. excelsa* reference genome[38] using bwa-mem2[73]. Although we aligned reads from multiple species (Dataset S1) against a single reference, alignment proportions were high for all samples (between 80 and 99%), likely reflecting the shallow divergence between the *Cochlearia* species[41]. We removed duplicated reads using Picard tools (https://broadinstitute.github.io/picard/) and identified SNPs using GATK4[74] (setting the appropriate -sample-ploidy option for each individual). Filtering of the variant calls was based on the GATK's best practices protocol, and we included filters for mapping quality (MQ ≥ 40 and MQRankSum ≥ −12.5), variant confidence (QD ≥ 2), strand bias (FS < 60), read position bias (ReadPosRankSum ≥ −8), and genotype quality (GQ ≥ 15). Following Monnahan et al.[21], we further removed SNPs with per-sample sequencing depth ≥ 1.6 × the mean depth to avoid issues caused by paralogous mapping.

### Analyses of genetic variation

We used the short-read-based SNP calls to infer genetic relationships among our diploid and polyploid *Cochlearia* populations. First, we estimated pairwise nucleotide diversity (π) and Tajima's *D* for each population using both mono- and biallelic sites. We then conducted a principal components analysis (PCA) using linkage-pruned ($r^2 \leq 0.1$ within 100 SNPs, minor allele frequency [MAF] > 0.05) SNPs found at synonymous (4-fold) sites. Following Patterson et al.[75], we estimated a covariance matrix representing the genetic relationships among each pair of individuals. For two individuals, *i* and *j*, covariance (*C*) was calculated as:

$$C_{ij} = \frac{1}{m} \sum_{s=1}^{m} \frac{(g_{is}/x_i - p_s)(g_{js}/x_j - p_s)}{p_s(1 - p_s)}, \tag{1}$$

where *m* is the number of variable sites, $g_{is}$ is the genotype of individual *i* in site *s*, *x* is the ploidal level of the individual, and *p* is the alternate allele frequency. We then conducted PCA on the matrix using the R function prcomp and extracted the first two axes of the rotated data for plotting. We also estimated genetic differentiation between populations using $F_{ST}$. Here, we employed the $F_{ST}$ measure by Hudson et al.[76], as recommended by Bhatia et al.[77].

To disentangle drivers of genetic differentiation among the *Cochlearia* populations, we tested for a pattern of isolation-by-distance and isolation-by-environment. Following Capblancq and Forester[63], we performed redundancy analyses (RDA) using the R package vegan[78]. We first estimated allele frequencies for the populations using linkage-pruned SNPs with MAF > 0.05 and ≤20% missing data. Missing population frequencies were imputed by randomly drawing them from a beta distribution with scale parameters calculated from the mean and variance of the non-missing values. We then extracted all 19 bioclimatic variables from WorldClim[79] and conducted forward model selection using RDA to identify a nonredundant set of variables explaining a significant proportion of genetic variation. Based on 1000 permutations, we kept 14 variables with *P* < 0.01. To transform the spatial structure of our data into a format usable in RDA, we conducted a principal coordinates analysis on a geographical distance matrix, retaining ten principal coordinates after forward model selection. Last, using partial RDA, we decomposed the effects of climate, geography, and ploidy in explaining genetic variation among the *Cochlearia* populations.

## Validation of the SV caller

We aligned both ONT and PacBio long-reads against the *C. excelsa* reference genome using minimap2[80] and identified SVs from the alignments using Sniffles2[46]. Our main analyses were based on 10 diploids (we excluded four diploids due to low sequencing depth, Supplementary Table 1) and seven tetraploids. As Sniffles2 expects the reads to originate from diploid organisms, we first used simulated data to evaluate its performance in autotetraploids. To estimate parameter values for the simulations, we used NanoPlot[81] to calculate the mean and SD of read lengths across all samples. By aligning reads from the *C. excelsa* reference individual against the reference genome, we estimated an empirical error rate of 4% for the ONT reads. We note, however, this is likely a conservative estimate, as we assumed that all differences between the assembly and sequencing reads were due to sequencing errors, whereas such differences may also result from misassemblies, erroneous alignments, or heterozygous SNPs (although heterozygous SNPs should be relatively rare in the selfing reference individual). We then used SURVIVOR[82] to simulate 10,000 random insertions and deletions between 50 bp and 100 kb into the *C. excelsa* reference genome. Using PBSIM2[83], we generated simulated ONT reads from the modified and unmodified FASTA files, and combined them assuming average read proportions for simplex (1/4), duplex (2/4), triplex (3/4), and quadruplex (4/4) mutations. The simulated read depth was either 5, 10, 20, 40, or 80. We last used minimap2 and Sniffles2 to conduct SV identification on the simulated data and calculated performance metrics (recall and precision) based on the results.

## Long-read-based SV identification

We called SV candidates individually for each sample and joined them into multi-sample VCF files using the population calling algorithm in Sniffles2[46]. To reduce false positives caused by misassemblies and erroneous alignments, we included the reference (highly homozygous) individual in all multi-sample VCF files and excluded SVs that were called heterozygotes or alternate homozygotes in the reference sample. We further focused our analyses on insertions and deletions (variant quality ≥ 20) between 50 bp and 100 kb, as methods based on read alignments are generally less accurate at detecting other types of SVs (e.g., tandem duplications and inversions) as well as very large SVs[58].

Although our simulations suggest that Sniffles2 has good power to detect insertions and deletions in autotetraploids (Supplementary Table 2), the genotype calls are incorrect due to the diploid-specific genotyping model. Therefore, we collected allele count data (i.e., the number of reads supporting the reference and alternate alleles in each variant) for SVs and used the R package Updog[84] to estimate genotype likelihoods and probabilities. We required that SVs used in Updog had ≤ 20% missing data and were covered by ≥ 10 reads in diploids and ≥ 20 reads in tetraploids. To include genotype uncertainty directly into our analyses, we estimated the allelic dosage, or the expected genotype, from the genotype probabilities as

$$E[G] = \sum_{g=0}^{4} g P(G = g), \tag{2}$$

where $G$ is the genotype. We then repeated this dosage estimation for the diploids to make the ploidy comparison equal.

## Differential methylation analysis

We assessed TE activity by quantifying differences in DNA methylation using our ONT sequenced samples (mean depth ≥ 10). To do so, we first used Tombo[85] to assign basecalls and genomic locations to raw signal reads. Then, based a model trained on *Arabidopsis thaliana* and *Oryza sativa* R9.4 reads, we used Deepsignal-plant[86] to estimate methylation frequencies in three sequence contexts, CG, CHG, and CHH (where H is A, T, or C). We last used cytosines covered by ≥ 6 reads to calculate methylation levels across TEs and genes.

To identify differentially methylated TE families between diploids and tetraploids, we used logistic regression and likelihood-ratio tests (LRTs) to search for associations between methylation levels and the ploidy. We controlled for the effects of population structure on methylation patterns by conducting a PCA on genome-wide methylation levels and including the supported number of PCs (defined using scree plots) as cofactors in the models. *P*-values from the LRTs were transformed to false discovery rate-based *Q*-values[87] to account for multiple testing. We considered TE families with $Q < 0.05$ as differentially methylated between the ploidies.

## Estimation of TE insertion times

We identified non-reference TE insertions from the ONT alignments using TELR, which has shown good performance in highly heterozygous, polyploid samples[88]. TELR combines Sniffles and RepeatMasker (http://www.repeatmasker.org) to first identify TE insertions and then performs local assembly of the inserted sequences using wtdbg2[89]. After running TELR on each ONT sequenced sample with mean depth ≥ 10, we aligned the inserted sequences against the consensus TEs using MAFFT[90] and calculated sequence divergence ($K$) using the F81 substitution model[91] implemented in the R package phangorn[92]. Last, we estimated insertion times using the following equation:

$$T = \frac{K}{2}/\mu, \tag{3}$$

where $\mu$ is the per year substitution rate, here assumed to be equal to the per-generation mutation rate estimated for *Arabidopsis thaliana* ($6.95 \times 10^{-9}$ per base pair[93]).

## Fitness effects of SVs

We assessed the fitness effects of SVs by first analyzing their allele frequency spectra (AFS). Using the estimates of allelic dosage, we calculated folded AFS for SVs found in exons, introns, ≤1 kb up and downstream of genes, and intergenic regions ( >1 kb away from genes). In the case of missing data (max 20%), we imputed the missing alleles by drawing them from a Bernoulli distribution. We further evaluated the selective removal of SVs by calculating the ratio of observed to expected numbers of SVs found overlapping the different genomic features (exons, introns, up and downstream, intergenic). The expected numbers were estimated by defining the proportion of the genome that is covered by each feature (i.e., under random expectations, SVs would be distributed according to those proportions). We note, however, that these expectations are likely affected by variation in mutation rates and insertion preference of TEs, but here were assumed that such biases are, on average, equal between the ploidies (this was confirmed for TEs, Supplementary Fig. 4–6).

To determine the level of selective constraint on genes affected by the SVs, we estimated coding sequence conservation using GERP + + [94]. We first selected 29 eudicot species from the clade Superrosidae (Supplementary Table 5), whose divergence times ranged from 20 million years (*Lobularia maritima*) to 123 million years (*Vitis vinifera*) in relation to *C. excelsa*[95]. To identify sequence homologs, we conducted BLAST searches against species-specific protein databases, selecting only the best match with an *e*-value $< 1 \times 10^{-5}$ for each gene. We aligned the coding sequences using MAFFT[90], keeping only homolog sets with 15 or more species. We then chose 1000 random genes with no missing species, extracted synonymous sites based on the *C. excelsa* sequence, and estimated a maximum likelihood tree using the R package phangorn[92]. Based on the species tree and multiple alignments, we used GERP + + to estimate the rejected substitutions score for sites in the *C. excelsa* coding sequence, indicating the degree of nucleotide conservation relative to the synonymous substitution rate. Finally, we

normalized the GERP scores using the range of possible values (as the range depends on the sample size of a particular site), calculated a median for each gene, and standardized the gene-specific estimates to a median of zero and MAD (median absolute deviation) of one.

## Pangenome construction and SV genotyping

To more broadly study the evolutionary impact of SVs in *Cochlearia*, we genotyped our long-read-based SVs in a set of 351 short-read sequenced individuals (Dataset S1). First, we identified SVs from all 23 long-read sequenced individuals, including four diploids previously excluded due to low sequencing depth, one hexaploid, and one octoploid, to construct a pangenome graph to serve as a reference for the short-read alignments. We kept all insertions and deletions filling the following requirements: not identified in the reference individual, length between 50 bp and 100 kb, variant quality ≥ 20, supported by ≥ 4 reads, and the proportion of supporting reads ≥ 0.1 of all reads. We then used vg[96] to construct a pangenome graph based on the chromosome-build *C. excelsa* reference genome[38] and the resulting 257,807 SVs. The short-read data were aligned to the pangenome graph using vg map[96] and SVs genotyped using vg call[97]. We last combined the individual-based SV calls into multi-sample VCF files using BCFtools[98] and estimated genotype probabilities and allelic dosage using Updog[84].

We further evaluated the performance of this genotyping pipeline using a similar approach as with the long-read data. First, we simulated 10,000 random insertions and deletions into the *C. excelsa* reference genome using SURVIVOR[82] and built a pangenome graph based on the simulations. We then masked half of the simulated SVs from the modified reference genome and generated simulated Illumina reads (paired-end, 150 bp) from the modified and unmodified FASTA files using Mason[99]. After aligning and genotyping SVs using vg, we calculated performance metrics based on the results. Note that by masking half of the simulated SVs, we were able to evaluate both recall and precision of the method (as vg only identifies SVs included in the graph). Analyses described in the following sections were conducted using SVs genotyped in the short-read sequenced samples.

## Genotype-environment association analyses

We tested for an association between genetic and environmental variables to identify loci potentially involved in local adaptation. To do so, we characterized the growing environment of 70 European *Cochlearia* populations (we excluded three populations from North America and three populations from Svalbard, as they represented clear climatic outliers) using 11 bioclimatic variables (Supplementary Fig. 11) identified with RDA (see "Analyses of genetic variation" for more details) and conducted genotype-environment association (GEA) analyses using BayPass[100]. BayPass was run on SV and SNP data compiled for three sets of samples: diploids, tetraploids, and all (diploids, tetraploids, and hexaploids combined). Note that BayPass works on population-specific allele frequencies (and not individual genotypes), making it suitable for polyploids. We required the variants to have MAF > 0.05 and ≤ 20% missing data to be included in the analyses. To control for the confounding effects of population structure, we included covariance matrices estimated using synonymous, linkage-pruned SNPs into all BayPass runs. Following the recommendation of Gautier[100], we repeated each run ten times with different seed numbers (settings for the priors and the MCMC sampling were left default) and calculated a median Bayes Factor (BF) for the variants. Variants with median deciban (dB) BF ≥ 10 were considered putatively adaptive (corresponding to strong evidence for an association between genetic and environmental variables).

## Analyses of candidate SVs and genes

To evaluate whether outlier SVs from our GEA analyses have been subject to recent positive selection, we used SweepFinder2[101] to scan areas around the SVs for signs of selective sweeps. We first chose populations with sample size ≥ 6 (12 diploid and 11 tetraploid populations) and then compiled SNP data from 20 kb regions around the breakpoints of each SV used in the GEA analyses. Using a custom grid search that included all variable sites within the 20 kb regions, we characterized the selective signals at each SV as the maximum composite likelihood ratio (CLR) test statistic found among the diploid or tetraploid populations (as local adaptation would not lead to sweep signals in all populations). For each population, we used the genome-wide site frequency spectrum (≤ 20 kb of SVs) as the neutral allele frequency distribution.

To better understand the functional importance of the outlier SVs, we conducted gene ontology (GO) enrichment analyses using the R package topGO[102]. For each outlier SV and SNP, we included the closest gene within 1 kb and ran GO enrichment analyses using the weight01 algorithm and Fisher's exact test. We defined the background distribution of GO terms using only genes ≤ 1 kb of SVs and SNPs. Following the recommendation of Alexa and Rahnenfuhrer[102], we considered GO terms with $P < 0.01$ as significantly enriched among the candidate gene sets.

## Climatic landscapes

We used RDA[78] to explore the climatic landscapes of SVs and SNPs. First, we estimated population allele frequencies for each climate-associated locus identified with BayPass (loci combined from diploid, tetraploid, and all runs) and imputed missing frequencies by drawing them from a beta distribution. We then used RDA to search for multivariate associations between allele frequencies and the 11 bioclimatic variables used in our GEA analyses. Following Capblancq and Forester[63], we used the loadings of the first two RDA axes to predict a climatic index for each environmental pixel across Europe (see Supplementary Fig. 12 for a biplot of the loadings). We next acquired occurrence data for our focal *Cochlearia* species from the Global Biodiversity Information Facility (GBIF)[103], and cleaned the records using an automated tool[104] and manual curation based on known *Cochlearia* growing sites and the GBIF photo gallery (see Supplementary Fig. 13 for a map of the occurrence records). We last summarized the results into 100 × 100 km grid points comprising the European range of the *Cochlearia* species. We did this prediction using both outlier SNPs and SVs, and plotted the climatic distance (Euclidean distance between the climatic indices) between the two variant types to identify geographical regions where SVs potentially make unique contributions to environmental adaptation. To explore possible effects of climate change on environmental adaptation, we used a Shared Socioeconomic Pathways (SSP) scenario SSP3-7.0[105] to model the increase in greenhouse gas concentrations by years 2061–2080.

## Reporting summary

Further information on research design is available in the Nature Portfolio Reporting Summary linked to this article.

## Data availability

Sequence data for this study have been deposited in the European Nucleotide Archive (ENA) at EMBL-EBI under accession number PRJEB66308. Source data are provided in this paper.

## Code availability

Scripts for conducting the analyses are available at GitHub [https://github.com/thamala/polySV][106].

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

## Acknowledgements

We thank S. Bray for assistance with sample collection, and J. Brookfield and A. MacColl for comments on the manuscript. We are grateful for access to the University of Nottingham's Deep Seq sequencing facility and Augusta HPC service. One collection required an explicit permit, which was granted by the Slovak Ministry for Environment (permission No. 062-219/18). The project has received funding from the European Union's Horizon 2020 research and innovation program under the Marie Skłodowska-Curie grant agreement No. 101022295 to T.H. and the European Research Council (ERC) grant agreements No. 679056 to L.Y., and No. 850852 to F.K. Funding was also received from the Leverhulme Trust under award No. RPG-2020-367 to L.Y.

## Author contributions

L.Y., T.H., and M.A.K. conceived the study. T.H. performed analyses. C.M., L.C., M.C., D.G., M.L., and L.Y. performed laboratory work. M.A.K., M.K.B., S.B., and F.K. provided materials. T.H. wrote the manuscript with input from other authors.

## Competing interests

The authors declare no competing interests.
