## [Peer Review File · Nature Communications]

Impact of whole-genome duplications on structural variant evolution in *Cochlearia* (Brassicaceae)Reviewers' Comments:

Reviewer #1:

Remarks to the Author:

The manuscript entitled "Impact of whole-genome duplications on structural variant evolution in the plant genus *Cochlearia*" presents a study to explore the impact of WGDs on SV landscapes as a function of ploidy, with a focus on understanding the influence of SVs on the adaptive potential of wild autopolyploid populations and species. The authors used a set of long-read sequenced samples from the plant genus *Cochlearia* (Brassicaceae), which contains diploids and a recent ploidy series up to octoploid.

The study of whole-genome duplication (WGD) has been a subject of significant scientific interest for decades, with numerous breakthroughs contributing to our understanding of these vital processes. While the methodology of the study may seem robust at first glance, closer examination reveals several critical issues that cast doubt on the validity of its findings.

The study employs a remarkably small sample size, with only a handful of plants in each treatment group. Small sample sizes can lead to skewed results, as random variations can have a disproportionate impact on the data. To draw meaningful conclusions, a more extensive and diverse set of plant samples should be used.

To mention more specific data, the authors only use 8 taxa of those currently accepted in the genus *Cochlearia* (see POWO, <https://powo.science.kew.org/taxon/urn:lsid:ipni.org:names:30020990-2>). The sampling conducted in this study appears to be insufficient (of the 76 localities, 23 of them have an $n < 3$, and the work is marred by significant taxonomic ambiguities. While the research effort is undoubtedly a step in the right direction, it falls short in providing a comprehensive understanding of the subject matter due to these limitations. Species, such as *Cochlearia anglica* L., *Cochlearia borzaeana* (Coman & Nyár.) Pobed., *Cochlearia danica* L., *Cochlearia gurulkanii* Yild., *Cochlearia* × *hollandica* Henrard, *Cochlearia micacea* E.S.Marshall, *Cochlearia scotica* Druce, *Cochlearia sessilifolia* Rollins, have not been taken into account in the general review of the genus, while others that were incorporated have not been adequately sampled as they do not represent the variability of their chromosome counts (as an example *Cochlearia groenlandica* L. and the situation in Iceland, see Olsen et al. 2022 or *Cochlearia officinalis* L. <https://www.tropicos.org/name/4100415>).

Furthermore, the work in question suffers from significant taxonomic ambiguities. Taxonomy is the science of classifying and naming living organisms, and it is imperative that this classification is accurate and unambiguous to ensure the validity of any ecological or biological study. Taxonomic ambiguities arise when there is uncertainty or confusion about the identification and classification of species, genera, or higher taxa. In this study, it is evident that taxonomic designations are either inconsistent, not adequately documented, or unclear, making it challenging to draw meaningful conclusions about the organisms involved. These ambiguities undermine the study's credibility and its potential to contribute to the broader body of scientific knowledge.

There are some problems in the hypothetical scenario of the evolution in *Cochlearia*. First, many described species are extremely similar to each other, and sometimes the boundaries between some of them are unclear (e.g. Koch et al. 1996). In fact, many species have been suspected to arise from hybrid speciation (e.g. Koch et al. 1998; Pegtel 1999; Lysak and Koch 2011). Additionally, polyploidization has also been an important speciation driving force in this tribe. Several studies devoted to this species group (Koch et al. 1996, 1998; Koch 2002) indicate a complex evolutionary history, including both auto- and allopolyploid origins (Marhold and Lihová 2006), showing a high morphological similarity among species, and phenotypic plasticity in different environments.

The authors assume different species without a clear criterion (e.g. *Cochlearia alpina*, *Cochlearia excelsa*, *Cochlearia pyrenaica*) when the main floras consider it as the taxon *Cochlearia pyrenaica* (see

POWO). There is a lack of support for relationships between different taxa of *Cochlearia* presented. To mention specific cases:

(1) *Cochlearia pyrenaica* was described by A.P. de Candolle (*Syst. Nat.*, Candolle 2: 365, 1821) from specimens collected by Ph.P. de Lapeyrouse (*Hist. Pl. Pyrénées*: 368, 1813) at different locations in the Pyrenees (Roche Saint-Vicent, Vicdessos, Glaciers d'Oo, M. d'Esquierry, des Cougous, Aiguecluse, Piquette d'Endretlis) under the name *C. officinalis*. Concerning *C. aestuaria*, it was described by J. Lloyd (*Fl. Ouest France, Herbor.* 3, 1880) from the French Atlantic coast as a variety of *C. officinalis* (i.e. *C. officinalis* var. *aestuaria*) and subsequently combined to species level by V.H. Heywood (*Feddes Reper.* 70: 6, 1965). In spite of the poorly defined morphological variability in the genus *Cochlearia* (see Koch et al. 1996, 1999), analysis of the ITS and plastid markers (*trnL* intron, *trnL-F*) suggests that the studied populations (including the locations indicated in the protologue) should be treated as *C. pyrenaica*. However, phylogenetic relationships between *C. pyrenaica* and *C. aestuaria* remain unclear, so it can be questioned whether they are two separate species or whether they should be united as one single species and then considered as two ecogeographical races of the same species. In view of the results based on ITS and *trnL-F* analysis, they should be classified at the subspecies level: an inland taxon *C. pyrenaica* subsp. *pyrenaica*, and a coastal taxon *C. pyrenaica* subsp. *aestuaria* (J.Lloyd) Fern.Casas & M.Laínz, an opinion already proposed by Fernández Casas (1975). This is also supported by isozyme analyses conducted by Koch et al. (1996), where fifteen out of sixteen alleles reported of *C. aestuaria* were shared with *C. pyrenaica*.

(2) *Cochlearia groenlandica* L. ($2n = 14$) (Gill 1971, 1973, 1976; Chater and Heywood 1964; Chater et al. 1993; Aiken et al. 1999). Additionally, other taxa with similar distribution (extreme northern America, Europe and Asia, including northern Japan) and hardly distinguishable from *C. groenlandica*, have also been described (Aiken et al. 1999; Al-Shehbaz and Koch 2010): *C. fenestrata* R.Br.; *C. polaris* Pobed.; *C. arctica* Schltld. ex DC., and *C. oblongifolia* DC. Indeed, chromosome counts indicate that these plants are $2n = 14$ (Aiken et al. 1999). *Cochlearia sessilifolia* Rollins is a North American plant without cytogenetic information, which, according to Al-Shehbaz and Koch (2010), should also be considered within the *C. groenlandica* group. As in the previous case, no chromosome counting was performed on *C. tridactylites* Banks ex DC. (= *Cochlearia cyclocarpa* S.F.Blake), another species from northeastern North America (Al-Shehbaz and Koch 2010).

(3) *Cochlearia danica* L. ($2n = 42$) (Gill 1976; Chater and Heywood 1964; Chater et al. 1993) is a plant distributed along the European Atlantic coasts.

(4) *Cochlearia tatrae* Borbas ($2n = 42$) (Gill 1976; Chater and Heywood 1964; Chater et al. 1993; Cieślak et al. 2007), an endemic plant of the western Carpathians (Tatra). Plants of section *Cochlearia* with basic chromosome number $x = 6$ are distributed in Europe between the Cantabrian Mountains and the Ukrainian Carpathians, reaching the northern edge of Scandinavia. This is a highly complex group with unclear systematics, within which many taxa have been described:

(5) *Cochlearia pyrenaica* DC. ($2n = 12$) (Gill 1971, 2007; Vogt 1985, 1987; Chater and Heywood 1964; Chater et al. 1993), a plant distributed in the Cantabrian Mountains and the Pyrenees, extending to the Ukrainian Carpathians. At the eastern end of the Alps (Austria), two closely related species have been described (*C. excelsa* Zahlbr. ex Fritsch and *C. macrorrhiza* Pobed.) with the same chromosome number ($2n = 12$).

(6) Likewise, plants very similar to *C. pyrenaica* grow in Britain, and have been described as *C. alpina* (Bab.) H.C.Watson [= *C. pyrenaica* subsp. *alpina* (Bab.) Dalby]. It is likely that these plants correspond to a tetraploid ($2n = 24$) (Gill 2007), although a diploid level ($2n = 12$) was occasionally reported in Britain (Gill 1971). However, this diploid level seems to correspond to plants of *C. pyrenaica* s.str. growing in the same territories.

(7) At the eastern end of the distribution area of *C. pyrenaica*, polyploid populations appear which

have been described as separate taxa: *C. polonica* A.Fröhl. ($2n = 36$) (Chater and Heywood 1964; Chater et al. 1993) in southern Poland; and *C. borzaeana* (Coman & Nyauady) Pobed. ($2n = 48$) (Chater et al. 1993; Cieślak et al. 2007; Kochjarová et al. 2006) in Romania.

(8) The origin of the plant in southern Germany described as *C. bavarica* Vogt ($2n = 36$) (Vogt 1985) seems to be the result of hybridization between *C. pyrenaica* and *C. officinalis* (Koch 2002).

(9) *Cochlearia aestuaria* (J.Lloyd) Heywood ($2n = 12$) (Gill 1971; Vogt 1987; Chater and Heywood 1964; Chater et al. 1993) is a plant that grows on the Atlantic coast of southwestern Europe, specifically in northern Spain, western France and southern Britain. Indeed, it is the only plant from this section with this chromosome number that exists along European coasts.

(10) *Cochlearia officinalis* L. ($2n = 24$) (Gill 1973, 2007) is a plant that grows spontaneously in many areas of the European Atlantic coast.

Plants with the same chromosome number and growing in northern coastal areas have been described as *C. scotica* Druce and *C. atlantica* Pobed. However, according to Gill (2007) this systematic treatment does not seem appropriate.

(11) *C. anglica* L. ($2n = 48$) (Chater and Heywood 1964; Chater et al. 1993; Gill 2007) grows in the middle latitudes of the Atlantic coast of Europe. *C. hollandica* Henrard ($2n = 36$) (Gill 1975) from the north Atlantic coasts, is interpreted as a hybrid between *C. officinalis* and *C. anglica*.

(12) *C. micacea* E.S.Marshall is a plant that grows in the mountains of northern Scotland. Recent data suggest that it is a hybrid originating from an ancient hybridization between plants from the *C. pyrenaica* group ($2n = 12$) and the *C. groenlandica* group ($2n = 14$). Indeed, *C. micacea* has a chromosome number of $2n = 26$ (Gill 1973, 2007).

Taking into account the review presented here, as well as the lack of support for relationships between different groups from the section *Cochlearia* the manuscript fails in adequate taxonomy and sampling, lack of analysis of possible appearance of hybrids and relationship of the samples. Many previous relevant publications are missing from the bibliography (e.g. Brandrud et al. 2017; Olsen et al. 2022, etc.). On the other hand, the study appears to be a standalone publication without any references to related research or comparative studies. This lack of context raises concerns about the potential for publication bias and the selective presentation of data to support a preconceived hypothesis.

Another worrying aspect is the acquisition of samples. The authors do not provide information on sampling locations, sampling permits since many species are protected taxa in different countries, herbarium sheets as a deposit reference, precise GPS coordinates, etc. The authors do not indicate how they obtained the samples. This aspect is very important under Nagoya protocol.

The Nagoya Protocol aims to provide legal certainty and transparency to both providers and users of genetic resources. This certainty is intended to promote responsible and sustainable utilization of these resources while ensuring that benefits are shared fairly. It establishes a clear legal framework for access to genetic resources, reducing ambiguity and the potential for biopiracy. The Nagoya Protocol serves as a vital international agreement to address the fair and equitable use of genetic resources and traditional knowledge. It represents a significant step toward promoting biodiversity conservation, respecting the rights of provider countries and communities, and ensuring that the benefits derived from genetic resources are shared fairly and transparently.

In the realm of scientific research, it is crucial to apply rigorous methodologies to ensure the reliability of findings. The scientific work on *Cochlearia*, while promising, suffers from several critical flaws that cast doubt on the validity of its conclusions. The absence of appropriate sampling, the small sample size, the taxonomic doubts, and the lack of important references to related research all contribute to the flawed approach in this investigation.

In conclusion, while the research in question may provide valuable insights, the sampling conducted is insufficient, and the presence of significant taxonomic ambiguities undermines the study's scientific rigor. Addressing these limitations is essential to ensure the accuracy and reliability of the findings and to contribute meaningfully to the field of taxonomy and ecology. Scientific research thrives when these issues are acknowledged and rectified, paving the way for more robust and conclusive investigations in the future.

Reviewer #2:

Remarks to the Author:

The manuscript presents an interesting study aiming to understand whether autopolyploidy results in greater diversity of genomic structural variants (SVs). The authors perform long-read sequencing of samples from the plant genus *Cochlearia* (Brassicaceae), which contains diploids and recent polyploids. SVs are found comparing sequencing reads to a diploid reference genome.

My main concern is relatively little SV validation, considering that some major analyses are based on rare SV calls.

Did I understand correctly that the simulations used to show performance of Sniffles2 on a polyploid genome, were based on a single locus?

From methods: 'We then randomly chose a single 10 kb region from the reference genome and introduced a 1 kb insertion or deletion into it. Using PBSIM279, we generated simulated ONT reads from the modified and unmodified FASTA files, and combined them assuming average read proportions for simplex (1/4), duplex (2/4), triplex (3/4), and quadruplex (4/4) mutations. The total read depth was either 5, 10, 20, 503 40, or 80 ´ the simulated region.'

This would at the very least need to be done on genome-wide level, for many SVs in different contexts, genic, non-genic etc, simulating realistic SV lengths and distributions. Performance of SV callers can vary depending on sequence context and SV properties.

Big part of the results and discussion is based on the observation that rare exonic SVs (singletons) are found in excess in tetraploids. I would still like to see PCR based validation of a random selection of the rare SVs observed in diploids and tetraploids.

There is no evaluation of the performance of a combination of vg map + vg call + updog for SV genotyping from population data with different ploidies. This would need to be done before the results can be used for any further analyses.

What do authors mean by ploidy specific SVs? These are SVs found only in diploids or tetraploids? But the geographic distribution (Fig 1A) and PCA analysis (Fig 1D) show strong geographic and genetic differentiation of diploids and tetraploids so the results can just reflect population structure/migration routes and not have anything to do with ploidy? Authors do aim to adjust for population structure, but I would like to see some additional evidence. For example, combining the SV analysis with phylogenetic analysis of diploids and tetraploids and showing that while SV is present in a tetraploid it is missing from the closest diploid relative. Do the ploidy specific SVs have a single origin or did they emerge independently multiple times?

Reviewer #3:

Remarks to the Author:

This manuscript details a study of population genetic variation in the polyploid plant genus *Cochlearia*, with a focus on structural variation. A limited series of diploid and autotetraploid individuals as well as a few individuals with higher ploidy (some from allo-polyploids) were sequenced using long reads (ONT or PACBIO HiFi), and a broader set was sequenced using Illumina short reads.

A PCA based on SNPs shows that ploidy is the main determinant of population subdivision, and polymorphism was lower in the polyploids than in the diploids. More structural variants (SV) were observed in polyploids than in diploids, with more singletons in the former. Several SV close to genes showed association with some environmental variables.

Overall, while I am convinced that the dataset is of potential interest, I also feel that the motivation of the work could be better presented, that several analyses remain too descriptive and that some conclusions are speculative.

1- the motivation for studying SV is not clearly explained, beyond the fact that it is now becoming technically possible. The analysis focuses on the efficacy of natural selection, and the correlation with environmental variables, but all of these aspects could already be addressed with SNPs. I feel that the current manuscript falls short of making clear what new elements we can expect to learn on the biology of polyploidy that was not accessible from SNPs already.

2- key elements of the evolutionary history of these diploid and polyploid lineages are missing from the current manuscript. In particular, if the polyploids correspond to a single origin (as the PCA suggests), then they do not represent evolutionary independent replicates, strongly limiting generality of conclusions on the impact of polyploidy. (I understand that these aspects are detailed in refs 40 and 42, but they should be clarified here).

3- The analysis of the site frequency spectrum is quite descriptive, and I am worried of several potential biases. a) First, the comparisons are made without any kind of statistical test, which are obviously necessary for any meaningful conclusion to be drawn. b) SNIFFLE and GATK make vastly different assumptions to call these two types of variation, each using their own set of arbitrary thresholds. How much they can be compared remains unclear to me, severely limiting the comparison that can be made between SNPs and SVs. The focus and reliance of the conclusions on singletons is particularly worrying here, since the SV vs SNP calling algorithms may be prone to different sources of bias for such rare variants. c) It would be interesting to mention whether the excess of singletons is also observed in SNPs, or whether that is specific to SVs. d) I am confused by the description of how the SFS were obtained. The methods section mentions a pangenome construction and SV genotyping approach, but the main text and Figure 2B seem to indicate a sample size of $n=10$ or $n=14$.

4- Big claims are made on the adaptive significance of the correlations between SVs and environmental variables, but at present I found the evidence presented to be weak. Testing whether these genomic regions carry signatures of recent selective sweeps could be a way to increase confidence in the significance of these correlations. Also, the results report the intriguing observation that fewer than expected genes and biological processes are pinpointed by both SNPs and SVs. This negative correlation is indeed unexpected and should at least be given a tentative explanation. Finally, given the very indirect nature of the evidence for the adaptive role of these variants, I found the conclusions drawn from the projection to future climates to be far-fetched and overly speculative. This part is also almost entirely disconnected from the main focus of the manuscript on polyploidy, so I would suggest to remove it.

Minor points:

- I am worried that the identification of SV using alignment to a common reference may create a reference bias. Would it be possible to de novo assemble the individual genomes instead ? Are the sequencing data of sufficient quality and depth for that ?
- the text mentions the intriguing observation that DNA methylation levels is higher in 4N than 2N accessions, but provides no explanation. Was this pattern observed before, and is expected ?
- would it be possible to use an outgroup to polarize the SNPs and SV, and unfold the SFS ?
- line 499 : what was this simulated 1kb insertion made of ? random sequence ? sequence sampled elsewhere in the genome ?

Reviewer #4:

Remarks to the Author:

I enjoyed reading this very well-written manuscript. In general, I think the analyses of SV in diploid and polyploid *Cochlearia* are clear, well-reasoned, and fairly interpreted. The only set of analyses that are somewhat concerning relate to the environmental aspects of the paper. I think the use of

occurrence records to gather climate data for individuals in relation to genomic features is interesting, but I think the authors go too far in referring to various climatic conditions as drivers and the patterns as adaptations. The results presented show associations of climate and SVs, nothing more. I therefore think that the language that indicates causation needs to be revised. I think a discussion point worth making is that this association may be more than that, but future work (and maybe spell out what this might be?) is needed to identify causation.

In addition, I have the following more minor comments.

1. Title. Perhaps give the family name in parentheses at the end?
2. Line 57. Regarding the point that doubling the genome leads to increased genetic diversity, this point was shown theoretically by Moody et al. (1993, Genetics) and empirically by Soltis and Soltis (1989 Evolution) and has been reported elsewhere by other authors as well.
3. Line 69. I suggest moving this clause to follow 'SVs' at the beginning of the sentence.
4. Line 73-74. See also papers by Dave Edwards and Jacqui Batley on crops and an additional paper on Amborella.
5. Line 79. Very interesting question!
6. Line 88. This point relates to my main concern above: it seems like these benefits would be difficult to document without long-term field experiments. Maybe the wording needs to be changed to be more associative than causative, with the idea of benefits presented as a hypothesis.
7. Line 100. How many species? And what is the overall distribution? More info on Cochlearia would be useful.
8. Line 101. Here and elsewhere: ploidy is a noun, and the adjectival form to modify levels is ploidal. Please correct throughout. (And yes, this is misused all the time.)
9. Line 104. Do you mean x here? How can the tetraploids have $n=6$? Please clarify.
10. Line 105. What are the chromosome numbers for the hexaploid and octoploid?
11. Lin 108. I agree – it would be sort of hard to sort out pangenomes if you don't know the genomic constitutions, right?
12. Line 117. Can you please clarify here? Do you mean polyploid populations? What about individual polyploids? Are the individuals more heterozygous or less or the same as diploids?
13. Line 120. Change to 'due to both'.
14. Line 125. Hyphenate 'between-population' as a compound adjective.
15. Line 158. Insert 'an' before 'equal'.
16. Line 217. This is where the issue of 'adaptation' arises; I think it would be much safer (and more appropriate) to say 'association' instead.
17. Line 240. Insert 'a' before 'larger'.
18. Line 245. Change 'covers' to 'cover' (data is plural).
19. Fig. 3b. What is the difference between All and Combined? Can this be clarified in the legend?
20. Line 303. Insert 'to their' before 'more'.
21. Line 306. I would say since its origin - it didn't exactly 'diverge' in a cladogenic way.
22. Line 314. Insert 'a' before 'population'.
23. Line 317 and surrounding area. This could all depend on how long the SVs are - short ones might not have much of an impact on meiosis, for example.
24. Same spot. Also, did you characterize genes in the pangenomes as 'core' and 'variable'? And if so, how did the variable genes vary with environment (as in crop plants per papers by Edwards, Batley, et al. and per Hodel et al. in Amborella, for example)?
25. Line 335. How do these genes relate to the associated environments?
26. Line 348 and surrounding area. It's not clear that it is the SVs, though, that are contributing – it is the entirety of the genotypes, which includes the SVs but also the allelic composition. If the SVs could be more specifically tied to traits and conditions, this argument could be strengthened considerably. As is, I think this can only be presented as a hypothesis, which would require toning down a lot of the 'adaptive' comments throughout. I very much like the direction this is going, but I don't think it's yet 'there' to make some of these conclusions.
27. Line 352 area. Try to make this area more specific.

28. Line 360. Please insert 'samples of varying ploidy from 2x to 8x' after 'on'.
29. Line 369. Insert 'an' before 'important'.
30. Line 411. What ploidy were the samples? Even though a table is cited, it might be nice to insert something about the range of ploidy included (e.g., 'representing diploids, tetraploids...').
31. Line 430. Change 'primarily' to 'primary'.
32. Line 432. Why is the tetraploid assembly smaller than the diploid? The assembly size and N50 stats need to be addressed in the Discussion.
33. Line 444. Low-quality.
34. Line 445. Clarify from which species.
35. Line 445. Reference
36. Line 453. Give rationale for this value.
37. Line 459. Autos can have more alleles at a site: an autotetraploid individual with tetrasomic inheritance can have 4 alleles at a single locus, etc.
38. Line 504. Insert 'a' before 'few'.
39. Line 523. Insert 'the' before 'diploid-specific'.
40. Line 533. How does this software work? And describe why the methylation analysis was done. It sort of comes out of the blue.
41. Line 551. Non-reference
42. Line 559. Per-generation
43. Line 595. Read-based
44. Line 598. Short-read should be short-read
45. Line 602. Insert 'of the' after 'build'.
46. Line 614. Change 'ran' to 'run'.
47. Line 635. Change 'loci' to 'locus' (it says 'each'; hence, the singular form is needed).
48. Line 641. How were these data records searched and downloaded? Were all Cochlearia records pooled, or were they downloaded by species? It could be difficult to obtain accurate locations if there are multiple cytotypes within species.
49. Line 642. Delete 'up'; 'cleaned' is sufficient.

Dear Reviewers,

We would like to thank you for your thoughtful assessments of our work. We have revised our manuscript in response to the constructive comments from all of you.

Please see our point-by-point responses below. Improvements made to the manuscript are highlighted.

Reviewer #2 (Remarks to the Author):

The manuscript presents an interesting study aiming to understand whether autopolyploidy results in greater diversity of genomic structural variants (SVs). The authors perform long-read sequencing of samples from the plant genus *Cochlearia* (Brassicaceae), which contains diploids and recent polyploids. SVs are found comparing sequencing reads to a diploid reference genome.

My main concern is relatively little SV validation, considering that some major analyses are based on rare SV calls.

Did I understand correctly that the simulations used to show performance of Sniffles2 on a polyploid genome, were based on a single locus?

From methods: ‘We then randomly chose a single 10 kb region from the reference genome and introduced a 1 kb insertion or deletion into it. Using PBSIM279, we generated simulated ONT reads from the modified and unmodified FASTA files, and combined them assuming average read proportions for simplex (1/4), duplex (2/4), triplex (3/4), and quadruplex (4/4) mutations. The total read depth was either 5, 10, 20, 50, 40, or 80 × the simulated region.’

1. This would at the very least need to be done on genome-wide level, for many SVs in different contexts, genic, non-genic etc, simulating realistic SV lengths and distributions. Performance of SV callers can vary depending on sequence context and SV properties.

***Thank you for the useful suggestion.** Yes, the previous validation was based on a single locus. We were interested in seeing how Sniffles deals with the different heterozygote classes present in tetraploids, and less on overall power (which was evaluated, e.g., in the Sniffles paper). We therefore chose an “easy” SV that likely would be found in diploids and checked how the method behaves in tetraploids.*

*That said, we **entirely agree with you** that it is possible that some combination of the genomic features and ploidy lead to unexpected complications. Therefore, we have now expanded our validation to include SVs introduced across the whole genome. With these data, we examined the power to detect SVs in genic and intergenic regions. (Table S2 and lines 551–558). Although we didn’t see evidence that certain genomic features are particularly problematic in polyploids, we did find that SV calls in the intergenic space (> 1 kb away from genes) may suffer from excessive rates of false-positives (~40%), likely because of high-density of repeats. By contrast, exons, intron, and 1 kb up- and downstream regions of genes all had low false-positive rates (~2%). These points are covered in lines 196–201.*

Therefore, we now present fully revised results (Fig. 2) and show that the excess of rare SVs in tetraploids is only found around exons. We further discovered that our previous approach of using a rare SV (or “non-germline”) mode in Sniffles led to worse results in genome-wide data. As a result, we have fully redone SV identification using the normal “germline” mode and repeated all analyses based on the new SV calls. The results stay largely unchanged, but thanks to this the SV data presented are now more robust.

2. Big part of the results and discussion is based on the observation that rare exonic SVs (singletons) are found in excess in tetraploids. I would still like to see PCR based validation of a random selection of the rare SVs observed in diploids and tetraploids.

As mentioned above, our new validation now includes SVs found overlapping different genomic features. From our simulations and empirical analyses we see no evidence that tetraploids would have higher false-positive rates at exonic regions than diploids, indicating that polysomic masking is the likely culprit for these patterns. We have also visually examined many long-read alignments and in all cases have confirmed the presence and approximate locations of the SV breakpoints. We therefore believe that PCR validation would add little to our manuscript while being a considerable time drain.

3. There is no evaluation of the performance of a combination of vg map + vg call + updog for SV genotyping from population data with different ploidies. This would need to be done before the results can be used for any further analyses.

*Our reasoning was that this genotyping approach is less affected by polyploidy than the de novo approach used with long-reads. This is because information about SV is reported regardless of the inferred genotype. Still, **we have now included a validation of the pipeline** in Table S3 (lines 651–660), confirming that it works well on polyploid samples (lines 249–251).*

4. What do authors mean by ploidy specific SVs? These are SVs found only in diploids or tetraploids? But the geographic distribution (Fig 1A) and PCA analysis (Fig 1D) show strong geographic and genetic differentiation of diploids and tetraploids so the results can just reflect population structure/migration routes and not have anything to do with ploidy? Authors do aim to adjust for population structure, but I would like to see some additional evidence. For example, combining the SV analysis with phylogenetic analysis of diploids and tetraploids and showing that while SV is present in a tetraploid it is missing from the closest diploid relative. Do the ploidy specific SVs have a single origin or did they emerge independently multiple times?

Yes, ploidy-specific means that an SV is present exclusively among diploid or tetraploid populations (line 296). As for the rest of the comment, we are not entirely sure what the reviewer means. Our sampling includes a broad representation of the Cochlearia genus, meaning that the closest diploid relative of the tetraploids is already included, and our analysis shows that the tetraploid-specific SVs are not found among these diploid species. This suggests that these SVs likely appeared after the two ploidal levels diverged. These ploidy-specific SVs very likely resulted from a single mutational event (as is true for any SV). Given the short time frame examined here (<300kyar; Wolf et al, 2021, ref 41 in the manuscript), it is especially unlikely that a second SV with near-identical breakpoints would have appeared independently.

Reviewer #3 (Remarks to the Author):

This manuscript details a study of population genetic variation in the polyploid plant genus *Cochlearia*, with a focus on structural variation. A limited series of diploid and autotetraploid individuals as well as a few individuals with higher ploidy (some from allo-polyploids) were sequenced using long reads (ONT or PACBIO HiFi), and a broader set was sequenced using Illumina short reads.

A PCA based on SNPs shows that ploidy is the main determinant of population subdivision, and polymorphism was lower in the polyploids than in the diploids. More structural variants (SV) were observed in polyploids than in diploids, with more singletons in the former. Several SV close to genes showed association with some environmental variables.

Overall, while I am convinced that the dataset is of potential interest, I also feel that the motivation of the work could be better presented, that several analyses remain too descriptive and that some conclusions are speculative.

Thanks for the encouraging comments. We now better point out the motivation for the work (point 1 below) and flag more explicitly where we speculate somewhat very late in the work.

1- the motivation for studying SV is not clearly explained, beyond the fact that it is now becoming technically possible. The analysis focuses on the efficacy of natural selection, and the correlation with environmental variables, but all of these aspects could already be addressed with SNPs. I feel that the current manuscript falls short of making clear what new elements we can expect to learn on the biology of polyploidy that was not accessible from SNPs already.

***Regarding motivation:** Thanks for pointing that out! It's true that **part** of the motivation for our study comes from the fact that accurate SV identification has become possible with the advent of long-read sequencing (as is true for any modern SV/pangenome study).*

***However, we have abundant specific motivation than this. We now better highlight this;** in the previous version it was only mentioned in the discussion. Now in the introduction, we say directly (lines 84-91):*

*“Given the increased mutational input, combined with more complicated recombination and DNA repair machinery², we may expect SV emergence to increase in polyploids compared to diploids. This hypothesis is supported by recent empirical work in both autopolyploid *Cochlearia officinalis*³⁷ and *Cardamine amara*³⁸, which point to rapid evolution of DNA repair genes. These selective sweeps suggest an early ‘mutator’ phenotype that generates excess SVs before the adaptation of the repair machinery to the polyploid cell state³⁷. Motivated by these theoretical and empirical results, we first quantify SV diversity in recent autopolyploids and then explore the evolutionary impact of the shifted SV landscape.”*

***Regarding the idea that ‘all of these aspects could already be addressed with SNPs’:** Once we established our results, in the discussion we note how SVs in particular can impact the evolutionary fate of polyploids (lines 349–357). Namely, that stronger negative fitness effects of SVs (compared to SNPs) would lead to a faster fitness decline of newly founded polyploid populations and the potentially faster generation rate of SVs in young polyploids would lead to greater reduction of fitness compared to SNPs. Both results contribute to the understanding of the evolutionary consequences of polyploidy – a topic that has been studied since the early 1900s, but only now are we the first to contrast in a system decoupled from hybridisation the effects of SV versus SNPs.*

*As for the genotype-environment associations, we **found that the majority of the candidate genes were not in fact detected using SNPs (lines 285–287), demonstrating that SVs can provide much novel information about climate-associated variation (lines 377–379).***

2- key elements of the evolutionary history of these diploid and polyploid lineages are missing from the current manuscript. In particular, if the polyploids correspond to a single origin (as the PCA suggests), then they do not represent evolutionary independent replicates, strongly limiting generality of conclusions on the impact of polyploidy. (I understand that these aspects are detailed in refs 40 and 42, but they should be clarified here).

Indeed, this PCA does not immediately show that all polyploids are likely of independent origin – likely a combination of shared ancestral polymorphisms and one or the other parent to be shared (in this case C. officinalis might have been involved multiple times) resulted in this pattern. In addition, all these polyploids (except tetraploid C. officinalis) are local endemics with limited genetic diversity compared to C. officinalis, which is among the most widespread taxa and consists also of various subspecies.

*We included the polyploid species C. bavaria, C. polonica, C. tatraea, C. officinalis and C. anglica. For most of these species we have detailed information about the evolutionary history and origin and all this information has been summarized and published recently by us (Wolf et al. 2021, eLife). **In appendix 1 of that publication, we gave a detailed description of any taxon/lineage including all relevant literature summarized.** In the same paper, there is a phylogenomic splits-tree network (Fig. 3, Wolf et al), which is better at sorting the different polyploids, and clearly showing independent evolutionary origin of those higher ploidy taxa. Tetraploid C. officinalis consists of a number of subspecies (mostly found in Scandinavia), but there are also two taxa from within the more broadly defined C. officinalis – namely C. micacea and C. alpina – which are found in the UK and distinct either cytogenetically (micaceae) and ecologically (alpina). In the UK flora, both are considered as taxa on species level, which we follow here, and for both taxa in particular cytogenetic differences also hint towards an independent origin of these two tetraploids compared to the common C. officinalis.*

We appreciate the comment of the reviewer, that in this respect our manuscript is in need of some clarification, and we added some of these aspects accordingly to the introduction (lines 93–101) and the results section (lines 123–131). Due to text limitations and because this paper should not be overloaded with taxonomy and systematics, these paragraphs have been kept short, referencing the appropriate studies for interested readers.

3- The analysis of the site frequency spectrum is quite descriptive, and I am worried of several potential biases. A) First, the comparisons are made without any kind of statistical test, which are obviously necessary for any meaningful conclusion to be drawn.

Conducting statistical testing on multiple AFS in a way that is not influenced by differences in sample sizes and/or the overall numbers of segregating sites is quite complicated and in our view without good precedent, especially in polyploids. To get around these issues, we efficiently summarised the AFS using Tajima's D and compared the overlap between 95% CIs to assess significance. We now make this point clear in the text (lines 203–206).

b) SNIFFLE and GATK make vastly different assumptions to call these two types of variation, each using their own set of arbitrary thresholds. How much they can be compared remains unclear to me, severely limiting the comparison that can be made between SNPs and SVs. The focus and reliance of the conclusions on singletons is particularly worrying here, since the SV vs SNP calling algorithms may be prone to different sources of bias for such rare variants.

Please see below.

c) It would be interesting to mention whether the excess of singletons is also observed in SNPs, or whether that is specific to SVs.

We completely agree that comparing SNPs and SVs is complicated by both technical and biological (e.g., mutation rate) differences. We have now emphasised that our main result is an excess of rare exonic SVs, which are directly affecting functional genomic elements. We don't see a similar excess at introns, up- and downstream regions, or intergenic space (although, as stated above, the intergenic SV calls may be less reliable) (Fig 2 B and C; lines 201–206). This combined with our greatly expanded validation (lines 196–201), makes this, in our opinion, strong empirical evidence of allelic masking in polyploids, as predicted by theory. We also don't see this excess of rare variants in SNPs (as shown in the earlier version of the manuscript). However, given both the technical and biological differences between SVs and SNPs, we have now decided to focus only on SVs in Fig. 2.

d) I am confused by the description of how the SFS were obtained. The methods section mentions a pangenome construction and SV genotyping approach, but the main text and Figure 2B seem to indicate a sample size of n=10 or n=14.

The AFS are conservatively based only on long-read data (10 diploids and 7 tetraploids). This is because the pangenome graph is used to genotype long-read based SVs in the short-read data, which biases the AFS and enrichment analyses in two ways: singletons in the short-read data are not true singletons, because they have already been found at least once in the long-read data, and the locations of the SVs are identical to those already detected in the long-read data. SV genotyping based on the pangenome graph is only used in the GEA analyses. This is mentioned in the results (line 247–249 and 291–292) and methods.

4- Big claims are made on the adaptive significance of the correlations between SVs and environmental variables, but at present I found the evidence presented to be weak. Testing whether these genomic regions carry signatures of recent selective sweeps could be a way to increase confidence in the significance of these correlations. Also, the results report the intriguing observation that fewer than expected genes and biological processes are pinpointed by both SNPs and SVs. This negative correlation is indeed unexpected and should at least be given a tentative explanation. Finally, given the very indirect nature of the evidence for the adaptive role of these variants, I found the conclusions drawn from the projection to future climates to be far-fetched and overly speculative. This part is also almost entirely disconnected from the main focus of the manuscript on polyploidy, so I would suggest to remove it.

Thank you for the useful suggestion. We have now tested for the presence of selective sweeps at areas around each SV (lines 682–692) and found that outliers in fact harbour, on average, stronger signals of sweeps than other SVs (Fig. 3B). We agree that the climate modelling is somewhat speculative and have toned down our interpretation of the results (also in response to comments made by reviewer #4). Some readers, however, may find this analysis useful (e.g., several readers of our preprint have expressed interest in this section), and we have therefore decided to keep it, while appropriately noting its speculative nature. For example, in this analysis we now refer to climatic landscapes (as opposed to adaptive landscapes), which better reflects the nature of the results.

Minor points:

- I am worried that the identification of SV using alignment to a common reference may create a reference bias. Would it be possible to de novo assemble the individual genomes instead? Are the sequencing data of sufficient quality and depth for that?

Reference bias is always a valid concern and avoiding it all together is likely not possible. Here, however, we think that reference bias is not a substantial issue due to the extremely weak species divergence. As evidence of this, we now added alignment rates to Table S1, which are very high in all samples (between 92 and 100%) and not different between the ploidies. As for assembling the genomes, this is complicated by the high heterozygosity of the polyploid genomes. This means that many or most heterozygous SVs (especially simplex mutations) would not be included in haploid representations of the de novo assembled

genomes, leading to a more biased and less sensitive set of SVs for the detection of low frequency variants.

- the text mentions the intriguing observation that DNA methylation levels is higher in 4N than 2N accessions, but provides no explanation. Was this pattern observed before, and is it expected ?

*We agree that this is indeed intriguing and perhaps grounds for a follow-up study. One explanation could be that our tetraploid populations are, on average, from higher latitudes than diploids. In *A. thaliana* (Kawakatsu et al. 2016 Cell) and *A. lyrata* (Hämälä et al. 2022 eLife) higher latitude populations tend to have higher methylation levels. However, we would prefer not to additionally speculate on this, because it's not important for the current study and would raise additional questions that are outside the core scope of this work.*

- would it be possible to use an outgroup to polarize the SNPs and SV, and unfold the SFS ?

*This would be especially problematic in *Cochlearia*, because there is no long-read data or assemblies available for the closest outgroup genus, *Ionopsidium* (9 M years diverged, Wolf et al. 2021). Other species with reference genomes available are already quite diverged, e.g. *Lobularia maritima* (20 M years diverged). In any case, we used the AFS analyses to characterise purifying selection, which is entirely appropriate from the folded spectra (and less susceptible to polarisation errors).*

- line 499 : what was this simulated 1kb insertion made of ? random sequence ? sequence sampled elsewhere in the genome ?

In the earlier simulations the insertion was a random, non-repetitive bit copied from elsewhere in the genome. As described above, we have now greatly expanded our validation approach to include SVs simulated across the genome.

Reviewer #4 (Remarks to the Author):

I enjoyed reading this very well-written manuscript. In general, I think the analyses of SV in diploid and polyploid *Cochlearia* are clear, well-reasoned, and fairly interpreted. The only set of analyses that are somewhat concerning relate to the environmental aspects of the paper. I think the use of occurrence records to gather climate data for individuals in relation to genomic features is interesting, but I think the authors go too far in referring to various climatic conditions as drivers and the patterns as adaptations. The results presented show associations of climate and SVs, nothing more. I therefore think that the language that indicates causation needs to be revised. I think a discussion point worth making is that this association may be more than that, but future work (and maybe spell out what this might be?) is needed to identify causation.

Thank you for your positive comments and thoughtful assessment of our work. To strengthen our analyses, we now show that climate-associated SVs carry stronger signals of selective sweeps than other SVs, suggestive of environmental adaptation. However, we recognise that neither climate-associations nor sweep signals are direct evidence of positive fitness effects and have also accordingly toned down our discussion and claims about the adaptive roles of SVs.

In addition, I have the following more minor comments.

1. Title. Perhaps give the family name in parentheses at the end?

Thank you; this is a good idea. Done.

2. Line 57. Regarding the point that doubling the genome leads to increased genetic diversity, this point was shown theoretically by Moody et al. (1993, Genetics) and empirically by Soltis and Soltis (1989 Evolution) and has been reported elsewhere by other authors as well.

Thanks, very relevant references – now cited.

3. Line 69. I suggest moving this clause to follow ‘SVs’ at the beginning of the sentence.

4. Line 73-74. See also papers by Dave Edwards and Jacqui Batley on crops and an additional paper on *Amborella*.

*Thanks for the suggestion. We now cite the *Amborella* paper.*

5. Line 79. Very interesting question!

Thank you. We think so too.

6. Line 88. This point relates to my main concern above: it seems like these benefits would be difficult to document without long-term field experiments. Maybe the wording needs to be changed to be more associative than causative, with the idea of benefits presented as a hypothesis.

We agree. As we mention above, we have now modified our wording throughout (highlighted in MS) to make it overall less causative. We did not, however, want to make it too ambiguous, given that environmental adaptation is the primary motivation underlying GEA analyses.

7. Line 100. How many species? And what is the overall distribution? More info on *Cochlearia* would be useful.

We fully agree that with the original version readers may want some more general information.

We therefore add key information about species number, distribution and ecology with the introduction (lines 93–101). Some other information is now also found with the first paragraph of the results section (lines 122-131).

8. Line 101. Here and elsewhere: ploidy is a noun, and the adjectival form to modify levels is ploidal. Please correct throughout. (And yes, this is misused all the time.)

Thanks for the point and the understanding/comment; we have now fixed this.

9. Line 104. Do you mean x here? How can the tetraploids have $n=6$? Please clarify.

N here refers to the base chromosome number (defined in line 118). Tetraploids have a base chromosome number $n = 6$ (now lines 122-123)

10. Line 105. What are the chromosome numbers for the hexaploid and octoploid?

We have now added the base chromosome numbers to the hexa- and octoploid species (lines 122–123).

11. Lin 108. I agree – it would be sort of hard to sort out pangenomes if you don't know the genomic constitutions, right?

*This is certainly more complicated among the higher ploidies, especially if (and likely when) they are a mixed auto- and allopolyploids. However, we know that the Cochlearia genomes are quite collinear, even between the different base chromosome numbers (our preliminary analysis suggests that the extra chromosome in $n = 7$ species is caused by a fission of one of the $n = 6$ chromosomes), so our read-alignment based approach should work well (e.g., the reads in the hexa- and octoploid samples align to the *C. excelsa* reference genome at very high percentages, Table S1).*

12. Line 117. Can you please clarify here? Do you mean polyploid populations? What about individual polyploids? Are the individuals more heterozygous or less or the same as diploids?

Thanks; yes, we meant polyploid populations (now fixed, line 138). Polyploid individuals are, on average, more heterozygous than diploids.

13. Line 120. Change to 'due to both'.

14. Line 125. Hyphenate 'between-population' as a compound adjective.

15. Line 158. Insert 'an' before 'equal'.

16. Line 217. This is where the issue of 'adaptation' arises; I think it would be much safer (and more appropriate) to say 'association' instead.

17. Line 240. Insert 'a' before 'larger'.

18. Line 245. Change 'covers' to 'cover' (data is plural).

All above in points 13-18 are now fixed. Thank you.

19. Fig. 3b. What is the difference between All and Combined? Can this be clarified in the legend?

All refers to a GEA analysis conducted on all populations, whereas combined was a set of candidate genes from diploid, tetraploid, and all GEA analyses. However, we have now reworked this part to focus on the results showing stronger evidence of positive selection and more detailed information about the candidate genes, and therefore have removed the GO enrichment test made on the combined candidate gene list.

20. Line 303. Insert ‘to their’ before ‘more’.
21. Line 306. I would say since its origin – it didn’t exactly ‘diverge’ in a cladogenic way.
22. Line 314. Insert ‘a’ before ‘population’.

All above in 20–22 are now fixed. Thanks.

23. Line 317 and surrounding area. This could all depend on how long the SVs are – short ones might not have much of an impact on meiosis, for example.

It is true that large SVs are more likely to influence recombination. However, here we see the clearest evidence of polysomic masking at exonic regions where even short SVs can have strong effects if they disrupt the coding sequence.

24. Same spot. Also, did you characterize genes in the pangenomes as ‘core’ and ‘variable’? And if so, how did the variable genes vary with environment (as in crop plants per papers by Edwards, Batley, et al. and per Hodel et al. in Amborella, for example)?

We have not characterised pangenomes based on their core and variable genes; this would only be accurate with highly complete de novo-assembled genomes of each taxon. We also feel that such analysis is not necessary for the current manuscript. We are in the process of such a large-scale genus wide T2T dataset generation, which is tremendously expensive and laborious; when ready, this will provide more suitable datasets for characterising the presence and absence of genes, so we prefer to perform this analysis more correctly in a follow-up work using the as-to-be-completed datasets.

25. Line 335. How do these genes relate to the associated environments?

Perhaps we misunderstood the question, but the association between allele frequencies and the climatic variation is statistical – we don’t know what the biological process is.

26. Line 348 and surrounding area. It’s not clear that it is the SVs, though, that are contributing – it is the entirety of the genotypes, which includes the SVs but also the allelic composition. If the SVs could be more specifically tied to traits and conditions, this argument could be strengthened considerably. As is, I think this can only be presented as a hypothesis, which would require toning down a lot of the ‘adaptive’ comments throughout. I very much like the direction this is going, but I don’t think it’s yet ‘there’ to make some of these conclusions.

Yes, thanks. This would still be the allelic composition of the SVs, unless we misunderstood the meaning of this comment. In the climatic landscape analysis, we compared the difference between SVs and SNPs, both of which were characterised on the same individuals. Certainly, this is not direct proof of adaptive effects (as is true for vast majority of population genomic studies of course), and we have accordingly toned down our discussion about the results, along with enhancing the analysis, as mentioned above.

27. Line 352 area. Try to make this area more specific.

We now made this point clearer.

28. Line 360. Please insert ‘samples of varying ploidy from 2x to 8x’ after ‘on’.
29. Line 369. Insert ‘an’ before ‘important’.
30. Line 411. What ploidy were the samples? Even though a table is cited, it might be nice to insert something about the range of ploidy included (e.g., ‘representing diploids, tetraploids...’).
31. Line 430. Change ‘primarily’ to ‘primary’.

Points 28-31 are now fixed. Thank you.

32. Line 432. Why is the tetraploid assembly smaller than the diploid? The assembly size and N50 stats need to be addressed in the Discussion.

This is the haploid genome size, which can be smaller in tetraploids due to genome fractionation. In fact, we previously showed that within Cochlearia there's a negative correlation between ploidy and DNA content per chromosome (Wolf et al. 2021, Fig. 1D). We also expect that purge_dups works less effectively on diploids than on tetraploids, based on our observations. However, given that these assemblies were only used to increase the diversity of annotated TE families, and were not analysed in any other way, we feel that discussing these points outside of methods would distract from the main point.

33. Line 444. Low-quality.

Fixed.

34. Line 445. Clarify from which species.

We now refer to the table listing each species (also listed in lines 119–122).

35. Line 445. Reference

Fixed

36. Line 453. Give rationale for this value.

Here we followed previous similar population genomic work (Monnahan et al 2019) where in our experience this value was found to provide protection against artefactual SNP calls while maintaining good genomic coverage.

37. Line 459. Autos can have more alleles at a site: an autotetraploid individual with tetrasomic inheritance can have 4 alleles at a single locus, etc.

Yes, this is true. Indeed, even a diploid population can have more than two alleles segregating in a single locus, but here we only focus on those loci that have two alleles, as is common practice in population genomic studies.

38. Line 504. Insert 'a' before 'few'.

39. Line 523. Insert 'the' before 'diploid-specific'.

Both are now fixed.

40. Line 533. How does this software work? And describe why the methylation analysis was done. It sort of comes out of the blue.

Thanks; we have now added a short explanation of why the methylation analysis was done (line 581–582).

41. Line 551. Non-reference

42. Line 559. Per-generation

43. Line 595. Read-based

44. Line 598. Short-real should be short-read

45. Line 602. Insert 'of the' after 'build'.
46. Line 614. Change 'ran' to 'run'.
47. Line 635. Change 'loci' to 'locus' (it says 'each'; hence, the singular form is needed).

Points 41-47 are now fixed. Thank you.

48. Line 641. How were these data records searched and downloaded? Were all Cochlearia records pooled, or were they downloaded by species? It could be difficult to obtain accurate locations if there are multiple cytotypes within species.

All Cochlearia species were downloaded together (by searching for "Cochlearia" in GBIF). There are probably some inaccuracies in the database, but as we focused on broad-scale distribution patterns across Europe, that is unlikely to influence our inferences. Furthermore, a huge number of populations (at several hundred) and regions have been visited by us throughout the last 30 years, which allowed to check and eliminate doubtful/wrongly annotated records (lines 709–710). The reviewer is right about between-species hybrids, which also occur if species grow in sympatry – however, these are found exclusively at coastal sites (C. officinalis, C. anglica, C. danica), so no novel (or incorrect) locations are included in the records because of those hybrids.

49. Line 642. Delete 'up'; 'cleaned' is sufficient.

Fixed.

Reviewer #1 (Remarks to the Author):

The manuscript entitled “Impact of whole-genome duplications on structural variant evolution in the plant genus *Cochlearia*” presents a study to explore the impact of WGDs on SV landscapes as a function of ploidy, with a focus on understanding the influence of SVs on the adaptive potential of wild autopolyploid populations and species. The authors used a set of long-read sequenced samples from the plant genus *Cochlearia* (Brassicaceae), which contains diploids and a recent ploidy series up to octoploid.

*We note early the idiosyncratic nature of this review, which is completely discordant with all three other reviewers, jumping to conclusions totally unshared by the three other reviews, and which we effectively rebut below. **Disturbingly, this review additionally misses basic aspects of the study, evidencing even that the reviewer did not read all the materials provided (see below).***

The study of whole-genome duplication (WGD) has been a subject of significant scientific interest for decades, with numerous breakthroughs contributing to our understanding of these vital processes. While the methodology of the study may seem robust at first glance, closer examination reveals several critical issues that cast doubt on the validity of its findings.

No work thus far has decoupled the effects of WGD from hybridisation to understand changes in the SV landscape upon WGD. We robustly address what this reviewer calls ‘critical issues’ below, and which we again emphasise none of the other three reviewers note (much less call ‘critical issues’) with our work.

1. The study employs a remarkably small sample size, with only a handful of plants in each treatment group. Small sample sizes can lead to skewed results, as random variations can have a disproportionate impact on the data. To draw meaningful conclusions, a more extensive and diverse set of plant samples should be used.

Our analyses were conducted at two levels: directly analysing SVs identified in long-read sequencing data and genotyping the long-read-based SVs in hundreds of sequence genomes using deep short reads.

We used these data primarily in two ways:

***First**, yes, the main objective was to assess shifts in SV spectra as a function of ploidy. Here, we used SVs identified directly in the long-read data to avoid biases associated with genotyping (see our response to reviewer #3’s comment 3d). Due to very high costs of large-scale long-read sequencing, the sample size is more limited (23 samples long-read sequenced deeply), but still comparable to other recent long-read/pangenome-based studies (e.g., Wang et al. 2023 NatComms, Weissensteiner et al. 2020 NatComms, Hufford et al. 2021 Science). That said, we assessed the fitness effects by analysing the AFS and enrichment of SVs found overlapping different genomic features. **For such analyses, power comes from the number of segregating variants, not from the total sample size.** Even in a more limited set of samples, the number of segregating SVs is high enough (> 100,000) for robust results with minimal level of uncertainty around the point estimates (small CIs in Fig. 2). Therefore, we are confident that the sample size is completely adequate for assessing the impact of WGDs on the SV landscape, as evidenced by our results.*

***Second**, we searched for SVs that potentially contributed to environmental adaptation by conducting genotype-environment association (GEA) analyses. **For these analyses, the sample size is more critical and we therefore strived for a broad sampling using short-read sequencing across the main *Cochlearia* range (351 samples from 76 populations).** As the GEA analyses are conducted on common SVs (MAF > 0.05), they are not influenced by the above-mentioned biases associated with genotyping. The reviewer specifically mentions (just below) populations that have $n < 3$. These populations were **used only in GEA analyses** (for the diversity estimates we required $n \geq 4$). For GEA the number of individuals per population is less important than the number of sampled populations (which we have a reasonable*

number). **Indeed, range-wide GEA is commonly conducted using only a single individual sampled from each population** (e.g., Rellstab et al. 2015 MER).

We now have added sample sizes in the abstract (lines 26 and 32) to make our sampling crystal clear.

2. To mention more specific data, the authors only use 8 taxa of those currently accepted in the genus *Cochlearia* (see POWO, <https://powo.science.kew.org/taxon/urn:lsid:ipni.org:names:30020990-2>). The sampling conducted in this study appears to be insufficient (of the 76 localities, 23 of them have an $n < 3$, and the work is marred by significant taxonomic ambiguities. While the research effort is undoubtedly a step in the right direction, it falls short in providing a comprehensive understanding of the subject matter due to these limitations. Species, such as *Cochlearia anglica* L., *Cochlearia borzaeana* (Coman & Nyár.) Pobed., *Cochlearia danica* L., *Cochlearia gurulkanii* Yild., *Cochlearia* × *hollandica* Henrard, *Cochlearia micacea* E.S.Marshall, *Cochlearia scotica* Druce, *Cochlearia sessilifolia* Rollins, have not been taken into account in the general review of the genus, while others that were incorporated have not been adequately sampled as they do not represent the variability of their chromosome counts (as an example *Cochlearia groenlandica* L. and the situation in Iceland, see Olsen et al. 2022 or *Cochlearia officinalis* L. <https://www.tropicos.org/name/4100415>).

*First, a little remark should be allowed herein due to an objective error in the above comment: *Cochlearia gurulkanii* Yild. does not belong to the genus *Cochlearia*, it belongs to the genus *Pseudosempervivum* and is been treated there correctly if the interested reader, e.g., consults BrassiBase (<https://brassibase.cos.uni-heidelberg.de/>) or relevant literature and not POWO.*

*While we indeed sample very broadly across *Cochlearia* diversity, this study never sets out to sequence the entire *Cochlearia* genus! Instead of reconstructing evolutionary history of the genus *Cochlearia*, our sampling aimed at covering the range-wide diversity in diploids and tetraploids in order to examine forces shaping potential differences in SV landscapes between these ploidal levels. We cover 13 of the 20 accepted taxa (the vast majority of the most frequent) and our field collections sample from an impressive 76 populations, which absolutely should not be called ‘insufficient’ or ‘marred’ for our core goal. **The reviewer erects a straw man: ‘a comprehensive understanding’ evidently of an entire plant genus: this is an artifactual projection by the reviewer, clear also below where we treat taxonomy.***

*We underscore that we used *Cochlearia* ploidy contrasts as a model for understanding how WGDs correspond to the evolutionary trajectory of SVs, as well as to scan for selective sweeps in SVs as a function of ploidy. This is **completely in-line, and indeed far surpassing** similar studies sampling broadly across ploidy contrasts to compare SNP-level variation (Monnahan et al. 2019 *NatEcolEvol*, Marburger et al. 2019 *NatComms*, Konečná et al. 2021 *NatComms*, Bohutínská et al. 2021 *MBE*, Bray et al. 2023 in revision for *Cell Reports*, also bioRxiv).*

*We note that we sample deliberately in order **to treat our research question regarding SV, and not attempting to comprehensively cover the entire genus, which should hardly be required in such a study!** For example, while *Cochlearia* is a good system for this work due to its ample ploidy variation and diverse environments inhabited by the species and populations, we have **deliberately excluded** a very common species, *C. danica*, because it is a highly selfing allohexaploid, which causes issues for both read alignments and biological interpretation of masking (due to high homozygosity and subgenome divergence).*

3. Furthermore, the work in question suffers from significant taxonomic ambiguities. Taxonomy is the science of classifying and naming living organisms, and it is imperative that this classification is accurate and unambiguous to ensure the validity of any ecological or biological study. Taxonomic ambiguities arise when there is uncertainty or confusion about the identification and classification of species, genera, or higher taxa. In this study, it is

evident that taxonomic designations are either inconsistent, not adequately documented, or unclear, making it challenging to draw meaningful conclusions about the organisms involved. These ambiguities undermine the study's credibility and its potential to contribute to the broader body of scientific knowledge.

We are very well aware indeed that the genus Cochlearia is a taxonomically complex and difficult group, and species recognition often followed various arguments – often not consistent – such as ploidal level, edaphic ecotypes, morphology and ecology. As a consequence, “names” (taxonomy) are manifold and most often do not reflect the evolutionary past. Phenotypic plasticity is high and often there is little consensus if morphological differences (in particular if it is quantitative variation) merits species and/or subspecies recognition. Taxonomic arguments (and expositions) can proceed ad nauseum, but here we focus on ploidy contrasts (across larger taxonomic units). While we certainly appreciate the fine-scale taxonomic details here enumerated, they are not in fact the topic of this MS.

That said, the reviewer should consult our (highly extensive, published) experience with Brassicaceae taxonomy and systematics over the last 30 years (e.g. Koch M.A., (...) and Franzke A. (2017) Database taxonomics as key to modern plant biology. Trends in Plant Sciences 23, 4-6, doi: 10.1016/j.tplants.2017.10.005.; Kiefer M., (...), Koch M.A. (2014) BrassiBase: Introduction to a Novel Knowledge Database on Brassicaceae Evolution. Plant Cell and Physiology 55 (1): e3.) dealing with the general issues of taxonomy in Brassicaceae as a whole.

***A first and reliable evolutionary characterization of Cochlearia has been presented by us in 2021** (Wolf E., (...), Yant L., Koch M.A. (2021) Evolutionary footprints of a cold relic in a rapidly warming world. eLife 10: e71572.). In the present manuscript, we built directly upon this evolutionary framework (Fig. 3 in Wolf et al. 2021) to **select carefully from the various evolutionary groups a representative set of diploids and polyploids** (again, not a comprehensive sequencing of the genus, **which is the straw man the reviewer here strangely belabours**). In the same paper, we critically summarized the various systematic-taxonomic aspects with an extensive appendix.*

Any accession studied is named with a reliable and proper taxonomic name allowing to link the name with appropriate information and metadata. As such: Cochlearia aestuaria, for example, is the only diploid in Salt Marshes. Cochlearia excelsa, for example, is the only diploid in alpine region on siliceous bedrocks. Cochlearia alpina, for example, is the UK inland stable tetraploid. Cochlearia micaceae, is the UK inland tetraploid showing respective chromosome number differences and a slightly different ecology. Cochlearia islandica is the 2n=12 from Iceland, whereas C. groenlandica is the arctic 2n=14. 2n=14 C. tridayctylitis has been shown to be set apart from other arctic 2n=14 Cochlearia (Wolf et al. 2021).

*To be specific on use of taxonomic levels: ***all our analyses* were conducted at the level of different ploidies. Indeed, the only section where species-level taxonomy was of impact was a PCA shown in Fig. S1, which in fact highlights that species assignments have little association with the genetic clustering of the populations.** Therefore, the shallow divergence and uncertain taxonomic assignments mentioned by the reviewer **are in fact an asset for our analysis**, as they are less likely to confound the effects of ploidy. We see no reason why the inclusion of additional arbitrary (and allopolyploid) species mentioned by the reviewer would change the signal of allelic masking that is the main result of our study (indeed, this study could have conducted only using C. pyrenaica and C. officinalis, which are the most common European diploid and tetraploid species in the genus, but we still strived for wider sampling).*

In summary, herein we present a very dense and representative sampling across the entire evolutionary space of diploid and polyploid taxa, consisting of 374 sequenced genomes– irrespectively of the various proposed taxonomic names. It should be also strongly underscored here, that any taxonomy used in the presented manuscript is correct.

There are some problems in the hypothetical scenario of the evolution in Cochlearia. First, many described species are extremely similar to each other, and sometimes the boundaries between some of them are unclear (e.g. Koch et al. 1996). In fact, many species have been suspected to arise from hybrid speciation (e.g. Koch et al. 1998; Pegtel 1999; Lysak and Koch 2011). Additionally, polyploidization has also been an important speciation driving force in this tribe. Several studies devoted to this species group (Koch et al. 1996, 1998; Koch 2002) indicate a complex evolutionary history, including both auto- and allopolyploid origins (Marhold and Lihová 2006), showing a high morphological similarity among species, and phenotypic plasticity in different environments.

The authors assume different species without a clear criterion (e.g. *Cochlearia alpina*, *Cochlearia excelsa*, *Cochlearia pyrenaica*) when the main floras consider it as the taxon *Cochlearia pyrenaica* (see POWO). There is a lack of support for relationships between different taxa of *Cochlearia* presented. To mention specific cases:

We treat this general point below (and above) this taxonomic enumeration from the reviewer. We note again that we are well aware of this taxonomic complexity, and appreciate the reviewer citing this work from our group extensively above, which should itself illustrate our good understanding of the system. By the way, it is also not true – just as another example – that C. excelsa is considered under a broadly defined C. pyrenaica. The relevant flora for this taxon is the Austrian Flora, and there it is considered as species. There is actually now accepted a subspecies concept of a broadly defined C. pyrenaica. And even then, taxonomic entities such as C. excelsa, C. aestuaria and C. macrorrhiza will “survive” taxonomically as distinct entities on subspecies rank.

(1) *Cochlearia pyrenaica* was described by A.P. de Candolle (Syst. Nat., Candolle 2: 365, 1821) from specimens collected by Ph.P. de Lapeyrouse (Hist. Pl. Pyrénées: 368, 1813) at different locations in the Pyrenees (Roche Saint-Vicent, Vicdessos, Glaciers d’Oo, M. d’Esquierry, des Cougous, Aiguecluse, Piquette d’Endretlis) under the name *C. officinalis*. Concerning *C. aestuaria*, it was described by J. Lloyd (Fl. Ouest France, Herbor. 3, 1880) from the French Atlantic coast as a variety of *C. officinalis* (i.e. *C. officinalis* var. *aestuaria*) and subsequently combined to species level by V.H. Heywood (Feddes Repert. 70: 6, 1965). In spite of the poorly defined morphological variability in the genus *Cochlearia* (see Koch et al. 1996, 1999), analysis of the ITS and plastid markers (trnL intron, trnL-F) suggests that the studied populations (including the locations indicated in the protologue) should be treated as *C. pyrenaica*. However, phylogenetic relationships between *C. pyrenaica* and *C. aestuaria* remain unclear, so it can be questioned whether they are two separate species or whether they should be united as one single species and then considered as two ecogeographical races of the same species. In view of the results based on ITS and trnL-F analysis, they should be classified at the subspecies level: an inland taxon *C. pyrenaica* subsp. *pyrenaica*, and a coastal taxon *C. pyrenaica* subsp. *aestuaria* (J.Lloyd) Fern.Casas & M.Lainz, an opinion already proposed by Fernández Casas (1975). This is also supported by isozyme analyses conducted by Koch et al. (1996), where fifteen out of sixteen alleles reported of *C. aestuaria* were shared with *C. pyrenaica*.

(2) *Cochlearia groenlandica* L. ($2n = 14$) (Gill 1971, 1973, 1976; Chater and Heywood 1964; Chater et al. 1993; Aiken et al. 1999). Additionally, other taxa with similar distribution (extreme northern America, Europe and Asia, including northern Japan) and hardly distinguishable from *C. groenlandica*, have also been described (Aiken et al. 1999; Al-Shehbaz and Koch 2010): *C. fenestrata* R.Br.; *C. polaris* Pobed.; *C. arctica* Schltld. ex DC., and *C. oblongifolia* DC. Indeed, chromosome counts indicate that these plants are $2n = 14$ (Aiken et al. 1999). *Cochlearia sessilifolia* Rollins is a North American plant without cytogenetic information, which, according to Al-Shehbaz and Koch (2010), should also be considered within the *C. groenlandica* group. As in the previous case, no chromosome counting was performed on *C. tridactylites* Banks ex DC. (= *Cochlearia cyclocarpa* S.F.Blake), another species from northeastern North America (Al-Shehbaz and Koch 2010).

(3) *Cochlearia danica* L. ($2n = 42$) (Gill 1976; Chater and Heywood 1964; Chater et al. 1993) is a plant distributed along the European Atlantic coasts.

(4) *Cochlearia tatrae* Borbas ($2n = 42$) (Gill 1976; Chater and Heywood 1964; Chater et al. 1993; Cieślak et al. 2007), an endemic plant of the western Carpathians (Tatra).

Plants of section *Cochlearia* with basic chromosome number $x = 6$ are distributed in Europe between the Cantabrian Mountains and the Ukrainian Carpathians, reaching the northern edge of Scandinavia. This is a highly complex group with unclear systematics, within which many taxa have been described:

(5) *Cochlearia pyrenaica* DC. ($2n = 12$) (Gill 1971, 2007; Vogt 1985, 1987; Chater and Heywood 1964; Chater et al. 1993), a plant distributed in the Cantabrian Mountains and the Pyrenees, extending to the Ukrainian Carpathians. At the eastern end of the Alps (Austria), two closely related species have been described (*C. excelsa* Zahlbr. ex Fritsch and *C. macrorrhiza* Pobed.) with the same chromosome number ($2n = 12$).

(6) Likewise, plants very similar to *C. pyrenaica* grow in Britain, and have been described as *C. alpina* (Bab.) H.C. Watson [= *C. pyrenaica* subsp. *alpina* (Bab.) Dalby]. It is likely that these plants correspond to a tetraploid ($2n = 24$) (Gill 2007), although a diploid level ($2n = 12$) was occasionally reported in Britain (Gill 1971). However, this diploid level seems to correspond to plants of *C. pyrenaica* s.str. growing in the same territories.

(7) At the eastern end of the distribution area of *C. pyrenaica*, polyploid populations appear which have been described as separate taxa: *C. polonica* A.Fröhl. ($2n = 36$) (Chater and Heywood 1964; Chater et al. 1993) in southern Poland; and *C. borzaeana* (Coman & Nyauady) Pobed. ($2n = 48$) (Chater et al. 1993; Cieślak et al. 2007; Kochjarová et al. 2006) in Romania.

(8) The origin of the plant in southern Germany described as *C. bavarica* Vogt ($2n = 36$) (Vogt 1985) seems to be the result of hybridization between *C. pyrenaica* and *C. officinalis* (Koch 2002).

(9) *Cochlearia aestuaria* (J.Lloyd) Heywood ($2n = 12$) (Gill 1971; Vogt 1987; Chater and Heywood 1964; Chater et al. 1993) is a plant that grows on the Atlantic coast of southwestern Europe, specifically in northern Spain, western France and southern Britain. Indeed, it is the only plant from this section with this chromosome number that exists along European coasts.

(10) *Cochlearia officinalis* L. ($2n = 24$) (Gill 1973, 2007) is a plant that grows spontaneously in many areas of the European Atlantic coast.

Plants with the same chromosome number and growing in northern coastal areas have been described as *C. scotica* Druce and *C. atlantica* Pobed. However, according to Gill (2007) this systematic treatment does not seem appropriate.

(11) *C. anglica* L. ($2n = 48$) (Chater and Heywood 1964; Chater et al. 1993; Gill 2007) grows in the middle latitudes of the Atlantic coast of Europe. *C. hollandica* Henrard ($2n = 36$) (Gill 1975) from the north Atlantic coasts, is interpreted as a hybrid between *C. officinalis* and *C. anglica*.

(12) *C. micacea* E.S.Marshall is a plant that grows in the mountains of northern Scotland. Recent data suggest that it is a hybrid originating from an ancient hybridization between plants from the *C. pyrenaica* group ($2n = 12$) and the *C. groenlandica* group ($2n = 14$). Indeed, *C. micacea* has a chromosome number of $2n = 26$ (Gill 1973, 2007).

Taking into account the review presented here, as well as the lack of support for relationships between different groups from the section *Cochlearia* the manuscript fails in adequate taxonomy and sampling, lack of analysis of possible appearance of hybrids and relationship of the samples. Many previous relevant publications are missing from the bibliography (e.g. Brandrud et al. 2017; Olsen et al. 2022, etc.). On the other hand, the study appears to be a standalone publication without any references to related research or comparative studies. This lack of context raises concerns about the potential for publication bias and the selective presentation of data to support a preconceived hypothesis.

We appreciate the taxonomic enumeration by the reviewer, with which we are in agreement (aside from the reliance on many antiquated markers e.g. in point 1: such markers are now very much outdated and very often fail to circumscribe species).

*We perform exactly such an exercise in a much more detailed and comprehensive form in our recent eLife study, which we clearly reference in this manuscript, along with appropriate botanical descriptions, exactly as above, **robustly refuting the reviewer's strange statement above that** our 'study appears to be a standalone publication without any references to related research or comparative studies'. In fact, to the best of our knowledge, we have cited all relevant papers addressing SV evolution in autopolyploids and we were not attempting to cite all previous studies conducted on the Cochlearia genus. For example, the two papers (Brandrud et al. 2017; Olsen et al. 2022) mentioned by the reviewer used low-density SNP data to study local adaptation and population structure in Nordic Cochlearia – neither of which was the topic of our study at all. We nevertheless now cite one these papers as part of our expanded introduction into the genus (line 97).*

Another worrying aspect is the acquisition of samples. The authors do not provide information on sampling locations, sampling permits since many species are protected taxa in different countries, herbarium sheets as a deposit reference, precise GPS coordinates, etc. The authors do not indicate how they obtained the samples. This aspect is very important under Nagoya protocol.

The Nagoya Protocol aims to provide legal certainty and transparency to both providers and users of genetic resources. This certainty is intended to promote responsible and sustainable utilization of these resources while ensuring that benefits are shared fairly. It establishes a clear legal framework for access to genetic resources, reducing ambiguity and the potential for biopiracy. The Nagoya Protocol serves as a vital international agreement to address the fair and equitable use of genetic resources and traditional knowledge. It represents a significant step toward promoting biodiversity conservation, respecting the rights of provider countries and communities, and ensuring that the benefits derived from genetic resources are shared fairly and transparently.

*The reviewer is **incorrect about all** of these points! **We gave detailed GPS coordinates** of all sampling locations in a supplementary Dataset S1 in the original submission. Evidently, also the reviewer neglected the reporting summary, **where we explicitly discuss that we gained permission from local authorities, appropriate permits, and permission from each country's Nagoya contacts** (as required), along with reporting due diligence to our local Nagoya Authority. This Reporting Information is published alongside articles in Nature Communications and is absolutely sufficient for reporting standards.*

We are in complete agreement that these issues are of paramount importance, and we now include an explicit section at the beginning of the methods (also immediately below) repeating this information we gave also in the Reporting Summary, detailing sampling for readers who neglect to read the supplementary tables or Nature Reporting Summary (which is published alongside the article).

“All samples were collected in compliance with local, national, and international laws in the following countries: Austria, Belgium, England, France, Germany, Iceland, Norway, Scotland, Slovakia, Spain, and Switzerland. Material from collections under curation/international exchange of Heidelberg botanical collections and herbarium (acronym HEID) was sourced ultimately between 2004 and 2022. Spatial scale represents the natural range of Cochlearia spp. Where applicable and relevant we received permission from Nagoya focal points in each country and submitted the Due Diligence Declaration to our relevant Competent Authority. No seed stocks were involved. Sampling of young leaf material into desiccant was performed in the field, aiming for at least 10 plants per population, with each sampled plant a minimum of 2 meters from any other. All locations are given in Dataset S1. Collection dates and locations are detailed

for all samples in the ENA archive at EMBL-EBI under accession number PRJEB66308 (<https://www.ebi.ac.uk/ena/browser/view/PRJEB66308>).”

In the realm of scientific research, it is crucial to apply rigorous methodologies to ensure the reliability of findings. The scientific work on *Cochlearia*, while promising, suffers from several critical flaws that cast doubt on the validity of its conclusions. The absence of appropriate sampling, the small sample size, the taxonomic doubts, and the lack of important references to related research all contribute to the flawed approach in this investigation.

In conclusion, while the research in question may provide valuable insights, the sampling conducted is insufficient, and the presence of significant taxonomic ambiguities undermines the study’s scientific rigor. Addressing these limitations is essential to ensure the accuracy and reliability of the findings and to contribute meaningfully to the field of taxonomy and ecology. Scientific research thrives when these issues are acknowledged and rectified, paving the way for more robust and conclusive investigations in the future.

We robustly refute these points above, point-by-point.

*We expect that the publicly available data generated in this study will be very useful for researchers interested in *Cochlearia* taxonomy and ecology, **but as stated above, our work is an evolutionary genomic study that addresses a completely different set of questions than a botanical survey.** We additionally note that this is primarily an evolutionary genomic study: the characterisation by this reviewer of it as ‘the field of taxonomy and ecology’ **betrays their forced mischaracterisation** (which no other reviewers share; indeed, all three of the others share none of this reviewer’s misapprehensions).*

*The major criticisms expressed by the reviewer are thus unfounded and none of the points raised about taxonomy, sample size, or citing previous research in *Cochlearia* cast doubt on our conclusions about the impact of whole-genome duplications on SV evolution, which are entirely adequately addressed in this unprecedented analysis.*

Finally, we appreciate that the expression of these concerns by the reviewer highlights that we could be clearer in the manuscript, which we have done throughout, as cited above.

Reviewers' Comments:

Reviewer #1:

Remarks to the Author:

I appreciate the authors' intention to refute each of the aspects raised in the previous review, many of which still do not convince me. Publishing in leading scientific journals implies first-class work. The authors defend that their study is solid in terms of the number of samples and analysis of the different taxa of the genus *Cochlearia* but at the same time they are surprised with phrases like "While we indeed sample very broadly across *Cochlearia* diversity, this study never sets out to sequence the entire *Cochlearia* genus! Instead of reconstructing the evolutionary history of the genus *Cochlearia*, our sampling aimed at covering the range-wide diversity in diploids and tetraploids to examine forces shaping potential differences in SV landscapes between these ploidal levels"

This study, in my point of view, should not be published because it is incomplete in terms of sampling and the number of species sampled. The authors are surprised by having to sample the entire genus when trying to justify that they have 13 of the 20 accepted taxa (just over 50%). The authors emphasize and they are aware of the difficulty of the group "We are very well aware indeed that the genus *Cochlearia* is a taxonomically complex and difficult group, and species recognition often followed various arguments – often not consistent – such as ploidal level, edaphic ecotypes, morphology and ecology. As a consequence, "names" (taxonomy) are manifold and most often do not reflect the evolutionary past. Phenotypic plasticity is high and often there is little consensus if morphological differences (in particular if it is a quantitative variation) merits species and/or subspecies recognition". However, given that the genus *Cochlearia* is a taxonomically complex and difficult group, and species recognition often followed various arguments, as the authors mention, such as the ploidal level, in addition to factors edaphic ecotypes, morphology and ecology, and that all of this has an impact on the names (taxonomy) and how to understand evolutionary processes from the past to the present, it is vitally important to have a representation of all accepted species to have a global vision or readers should wait for a second publication within a couple of years to decipher the enigma? In conclusion, I do not recommend its publication.

Reviewer #2:

Remarks to the Author:

The authors provided a comprehensive answer to my queries and I am satisfied with their responses.

Reviewer #4:

Remarks to the Author:

I think the authors have done an excellent job of responding to the reviewers' comments. Although I might not agree with all of the responses or the approaches used to address the comments, I appreciate the conscientious approach to the reviews and the extensive additional work by the authors, and I recognize that there are multiple ways to address scientific questions. I have reviewed the reviews, the response to the reviews, and the highlighted changes to the ms in the context of the entire ms and have only a few minor comments. Although not an expert on certain aspects of the genomic data analysis, I think the authors' approaches to validating the methods and SV detection strengthen the paper. I will leave further assessment of these methodological revisions to Rev 2 and 3. I appreciate the change from 'adaptive landscape' to 'climatic landscape' (but note that Fig S7 still uses 'adaptive index' where perhaps 'climatic index' would be more consistent with the revised terminology).

1. Line 35. Perhaps insert 'possible' before 'local adaptation'? As addressed previously, there is no evidence of adaptation (although the patterns are certainly suggestive).
2. Lines 121-123. I think x is the appropriate symbol for base number and n is the appropriate signal

for the gametic (haploid) number, such that a tetraploid would have $n = 12$ if $x = 6$ and $n = 14$ if $x = 7$. I really think this should be changed.

3. Line 201. I would remove 'with' at the end of the sentence to avoid ending the sentence in a preposition. I think the sentence can end with overlapped. Or change to 'regardless of the elements with which the SVs overlapped', but I think ending with overlapped is fine.

4. Line 244. I like the change to 'climate-associated SVs' but note that a hyphen is needed (compound adjective).

5. Line 259. Statistics rather than statics

6. Line 264. Wilcoxon rather than Wilcox

7. Line 302. I would insert 'may' before 'make'. I don't think this weakens your study – it is just more honest about the current status of this discussion.

8. Line 653. SURVIVOR

9. I wonder if it would be useful to remove correlated bioclimatic variables prior to the RDA and other analyses.

10. Finally, I appreciate that the paper is on evolutionary genomics and is a comparison between diploids and tetraploids, not a paper designed to sort out relationships within Cochlearia, but taxonomy IS incredibly important, especially when integrating data across sources, such as the authors' genomic data with occurrence records on GBIF. In response to Rev 4, the authors indicated that there could be few errors in which records of one taxon were included in a different taxon. However, given the taxonomic past described by Rev 1 (and which the authors have apparently helped to straighten out), the possibility of outdated taxonomy on specimen records (and therefore the inclusion of those records in the 'wrong' taxon) seems potentially high. In my own work, different taxon concepts in a group can prove very challenging when trying to assemble a data set from GBIF because only rarely do specimens have updated names applied to them; thus, a data set may (likely?) include specimens from what may now be multiple taxa but that were at one time or another included under the same name. It might be worth some additional validation by checking a random set of images for each species to ensure the specimens represent what the authors think/hope they represent.

Reviewers' comments (in black) and our replies (in blue)

Reviewer #1 (Remarks to the Author):

I appreciate the authors' intention to refute each of the aspects raised in the previous review, many of which still do not convince me. Publishing in leading scientific journals implies first-class work. The authors defend that their study is solid in terms of the number of samples and analysis of the different taxa of the genus *Cochlearia* but at the same time they are surprised with phrases like "While we indeed sample very broadly across *Cochlearia* diversity, this study never sets out to sequence the entire *Cochlearia* genus! Instead of reconstructing the evolutionary history of the genus *Cochlearia*, our sampling aimed at covering the range-wide diversity in diploids and tetraploids to examine forces shaping potential differences in SV landscapes between these ploidal levels"

This study, in my point of view, should not be published because it is incomplete in terms of sampling and the number of species sampled. The authors are surprised by having to sample the entire genus when trying to justify that they have 13 of the 20 accepted taxa (just over 50%). The authors emphasize and they are aware of the difficulty of the group "We are very well aware indeed that the genus *Cochlearia* is a taxonomically complex and difficult group, and species recognition often followed various arguments – often not consistent – such as ploidal level, edaphic ecotypes, morphology and ecology. As a consequence, "names" (taxonomy) are manifold and most often do not reflect the evolutionary past. Phenotypic plasticity is high and often there is little consensus if morphological differences (in particular if it is a quantitative variation) merits species and/or subspecies recognition". However, given that the genus *Cochlearia* is a taxonomically complex and difficult group, and species recognition often followed various arguments, as the authors mention, such as the ploidal level, in addition to factors edaphic ecotypes, morphology and ecology, and that all of this has an impact on the names (taxonomy) and how to understand evolutionary processes from the past to the present, it is vitally important to have a representation of all accepted species to have a global vision or readers should wait for a second publication within a couple of years to decipher the enigma?

In conclusion, I do not recommend its publication.

We again appreciate the reviewer's interest in the *Cochlearia* genus and agree that having representatives of every species in a genus is ideal for studies of taxonomy and systematics. However, as we already explained in detail in our previous rebuttal, our work is an evolutionary genomics study focused on understanding the impact of WGDs on the evolutionary trajectory of SVs.

To study this without the confounding effects of hybridisation (as explained in the manuscript), we have focused on comparing diploid and **autopolyploid** *Cochlearia* species. Of the seven taxa not included in our study, all polyploids are likely of **allopolyploid** origin (Wolf et al. 2021 *eLife*). Therefore, including those species would not help us to understand the influence of WGDs on SV evolution, but rather confound it. All this is clearly covered in the manuscript and in our previous rebuttal. Furthermore, we strongly believe that sequencing the entire genus should not be a requisite of publishing any scientific paper, even in the top journals (indeed, vastly few papers published in *Nature Communications* fill such a requirement). Although we focus on a specific evolutionary process, our sampling across the *Cochlearia* genus is remarkably extensive, covering 13 of 20 accepted taxa, only leaving out very rare or otherwise problematic (*C. danica*, which a highly selfing **allopolyploid**) species. In sum, the reviewer's demand to include the whole *Cochlearia* genus into our study is not only unreasonable but also useless, as it would not help us to better understand in the impact of WGDs on SV evolution.

Reviewer #4 (Remarks to the Author):

I think the authors have done an excellent job of responding to the reviewers' comments. Although I might not agree with all of the responses or the approaches used to address the comments, I appreciate the conscientious approach to the reviews and the extensive additional work by the authors, and I recognize that there are multiple ways to address scientific questions. I have reviewed the reviews, the response to the reviews, and the highlighted changes to the ms in the context of the entire ms and have only a few minor comments. Although not an expert on certain aspects of the genomic data analysis, I think the authors' approaches to validating the methods and SV detection strengthen the paper. I will leave further assessment of these methodological revisions to Rev 2 and 3. I appreciate the change from 'adaptive landscape' to 'climatic landscape' (but note that Fig S7 still uses 'adaptive index' where perhaps 'climatic index' would be more consistent with the revised terminology).

Thank you for encouraging comments (and the legend title in Fig. S7 is now fixed).

1. Line 35. Perhaps insert 'possible' before 'local adaptation'? As addressed previously, there is no evidence of adaptation (although the patterns are certainly suggestive).

Thanks, that's a fair point. That's changed.

2. Lines 121-123. I think x is the appropriate symbol for base number and n is the appropriate signal for the gametic (haploid) number, such that a tetraploid would have $n = 12$ if $x = 6$ and $n = 14$ if $x = 7$. I really think this should be changed.

Thanks. We now use x to refer to the base chromosome number.

3. Line 201. I would remove 'with' at the end of the sentence to avoid ending the sentence in a preposition. I think the sentence can end with overlapped. Or change to 'regardless of the elements with which the SVs overlapped', but I think ending with overlapped is fine.

Thanks. The sentence is fixed.

4. Line 244. I like the change to 'climate-associated SVs' but note that a hyphen is needed (compound adjective).

5. Line 259. Statistics rather than statics

6. Line 264. Wilcoxon rather than Wilcox

7. Line 302. I would insert 'may' before 'make'. I don't think this weakens your study – it is just more honest about the current status of this discussion.

8. Line 653. SURVIVOR

All the above are fixed.

9. I wonder if it would be useful to remove correlated bioclimatic variables prior to the RDA and other analyses.

This is indeed common practise, but here we used forward model selection to choose explanatory variables, which, we believe, is a more sophisticated approach. Forward selection starts from a null model where the response is explained only by an intercept. Variables are then added to the model one by one to try to reach the amount of variance explained by the full model (i.e., model including all the explanatory variables), while limiting the amount of redundancy among included variables (now mentioned in line 532). Therefore, this approach gives a nonredundant set of explanatory variables while avoiding the use of arbitrary cut-offs for the correlation coefficient.

10. Finally, I appreciate that the paper is on evolutionary genomics and is a comparison between diploids and tetraploids, not a paper designed to sort out relationships within Cochlearia, but taxonomy IS incredibly important, especially when integrating data across sources, such as the authors' genomic data with occurrence records on GBIF. In response to Rev 4, the authors indicated that there could be few errors in which records of one taxon were included in a different taxon. However, given the taxonomic past described by Rev 1 (and which the authors have apparently helped to straighten out), the possibility of outdated taxonomy on specimen records (and therefore the inclusion of those records in the 'wrong' taxon) seems potentially high. In my own work, different taxon concepts in a group can prove very challenging when trying to assemble a data set from GBIF because only rarely do specimens have updated names applied to them; thus, a data set may (likely?) include specimens from what may now be multiple taxa but that were at one time or another included under the same name. It might be worth some additional validation by checking a random set of images for each species to ensure the specimens represent what the authors think/hope they represent.

We fully agree that taxonomy is important and that the complexity of Cochlearia could introduce bias into some of our analyses. As a result, we have now conducted more careful curation of the GBIF occurrence records.

First, we previously included the whole Cochlearia genus into the GBIF-based analyses, but to make curation more feasible, we now focus only on the species included in our study (making also the inference more valid). Consequently, many of the inland areas are excluded due to removal of the invasive, highly abundant *C. danica* (not included in the study due to being highly selfing allopolyploid), which likely has different adaptive requirements to our focal species.

Second, we inspected hundreds of plant photos available in GBIF and confirmed that they appear, in our experience, to depict the correct species. However, most occurrence records are not associated with photos, so we think that this approach is ultimately of limited value. Nevertheless, given the more limited set of occurrence records now included, we were able to use our own sampling (Fig. A1) and extensive experience in Cochlearia (Koch et al. 1992, 1998, 2003; Koch 1999, 2002, 2012; Wolf et al. 2021; Bray et al. 2023) to manually curate the records and create accurate continent-level ranges for the Cochlearia species (shown in Figs. 4, S7, and S8; the curated occurrence records are shown in Fig. S13).

The reviewer is right in pointing out that old/different naming conventions do influence the GBIF records and as part of our manual curation we have fixed many of these (for example, *C. islandica* is considered as subspecies of *C. officinalis* in GBIF even though they have different ploidal levels!). In any case, as the GBIF records were only used to define the range in certain maps (Figs. 4, S7, S8), any remaining uncertainty about Cochlearia subspecies is not a substantive issue, because the associated grid points (100 x 100 km) would be included regardless of which species the records come from as long as the species is included in the study.

In the previous review the reviewer also pointed out possible hybrids between the Cochlearia species. These hybrids, as mentioned in our response, occur exclusively in the coastal areas. As seen from Fig. S13, these regions are also heavily occupied by the pure species, so no incorrect grid points are included due to those hybrids. Furthermore, before progressing with our analyses, we have already excluded sequenced individuals that were suspected hybrids or had uncertain species/ploidy assignment (we originally sequenced over 400 individuals).

Reviewers' Comments:

Reviewer #4:

Remarks to the Author:

Despite the comments of Reviewer 1, I think the authors have adequately addressed the vast majority of the criticisms through multiple rounds of review. I have reviewed their responses as well as their revisions of the ms, and I recommend that the ms be accepted for publication.